# Cognitive and neural mechanisms of mental imagery supporting creative cognition
Jing Gu [1,2], Xueyang Wang[1,2], Cheng Liu[1,2], Lin Yang [1,2], Jiaxin Fan[1,2], Jiangzhou Sun [3 ✉], Yoed Nissan Kenett [4 ✉] & Jiang Qiu [1,2,5 ✉]

While the role of mental imagery in creative cognition is acknowledged, the specific cognitive and neural mechanisms remain underexplored. This study aims to elucidate the supportive role of mental imagery in creative cognition from a semantic memory perspective, and elucidating its underlying neural substrates. Initially, we conducted a behavioral study and found positive correlation between the vividness of touch imagery with creative performance in a creative writing task. By establishing semantic feature indicators based on writing texts and mediation models, we found that the vividness of touch imagery facilitates creative writing performance by semantic integration and reorganization. A subsequent behavioral study comparing mental imagery and semantic understanding strategies usage in creative writing tasks further confirmed the positive impact of mental imagery on creative cognition, and suggested that semantic reorganization, beyond the role of semantic integration, plays a critical role in how mental imagery enhances creativity. Finally, a functional magnetic resonance imaging (fMRI) study explored the distribution of functional brain networks' edge communities during creative writing under mental imagery and semantic understanding conditions. We found that the sensorimotor network facilitates sensorimotor simulations in creative cognition; the dorsal attention and salience networks collaboratively support the writing process by maintaining goal-directed attention and reorienting attention; the limbic network supports multimodal semantic processing and novel associations; the frontoparietal control network and default mode network contribute to information integration; and a subnetwork of default mode network plays a special role in integrating semantic information related to objects and actions. Collectively, our study sheds light on the cognitive and neural underpinnings of mental imagery supporting creative cognition.

Mental imagery, a fundamental ability enabling us to plan, rehearse future events, re-examine the past, and even simulate unreal scenarios[1], has been central to discussions of mental function for thousands of years, from philosophers to psychologists and now neuroscientists. Mental imagery is an information processing process that occurs in working memory[2]. It involves retrieving sensory attributes stored in long-term memory and constructing neural representations that resemble the results of perceptual processes[3]. Such images can reflect veridical memories (e.g., a loved one's face) or fantastical constructs (e.g., a green elephant)[4]. Some of the best

anecdotal accounts of imagery facilitating imaginative thought come from scientists. Albert Einstein described his thought process as the involvement of the voluntary reproduction and combination of clear images[5]. This one example highlights the importance of mental imagery in fostering high-order cognition, such as creative cognition[6].

Creative cognition involves the integration of multiple cognitive processes[7–9], including idea generation and idea evaluation processes[7,10,11]. Einstein's description is supported by findings that explicit, conscious imagery predicts creative performance[12]. While some recent research has

[1]Key Laboratory of Cognition and Personality (SWU), Ministry of Education, Chongqing, China. [2]Faculty of Psychology, Southwest University, Chongqing, China. [3]College of International Studies, Southwest University, Chongqing, China. [4]Faculty of Data and Decision Sciences, Technion, Israel Institute of Technology, Kiryat Hatechnion, Haifa, Israel. [5]West China Institute of Children's Brain and Cognition, Chongqing University of Education, Chongqing, China. ✉e-mail: sunjiangzhou@swu.edu.cn; yoedk@technion.ac.il; qiuj318@swu.edu.cn

acknowledged its role in creativity[13–17], its underlying mechanisms remain unclear. Here, we aim to explore how mental imagery supports creative cognition, particularly in creative writing, at the cognitive and neural levels.

This study investigates creative cognition via creative writing. The creative cognition approach views creativity as the generation of novel and appropriate products through the application of basic cognitive processes to existing knowledge structures[18]. Ward[19] established writing as a behavioral domain to study creative cognition, demonstrating how narrative construction inherently relies on cognitive processes like conceptual combination. Unlike traditional creative divergent or convergent thinking tasks, creative writing entails a more intricate creative cognition process and reflects everyday creativity (creative hobbies, problem-solving in leisure or work activities)[20–23]. Research has increasingly positioned creative writing as a behavioral manifestation of creative cognition. For instance, Taylor and Barbot[24] demonstrated that the dual pathways of creativity (executive control and associative abilities) are directly applicable to creative writing productions, while studies on writing subprocesses reveal that optimal creativity emerges from dynamic patterns of idea generation and selection[25]. Notably, the sustained nature of creative writing could help to reveal the involvement of broader cognitive processes in creative cognition. Studies indicate that extended engagement with semantically complex and coherent narratives—as opposed to fragmented stimuli—recruits higher-order brain networks such as the default mode network[26], suggesting that creative writing involves complex cognitive processes. Collectively, creative writing is a feasible way to reveal the involvement of broader cognitive processes in creative cognition.

Creative cognition is fundamentally embodied, with mental imagery serving as a critical cognitive process between sensorimotor experiences and creative outputs. This relationship manifests through two complementary research trajectories. First, embodied cognition studies highlight the scaffolding role of physicality in creative processes. Body movements and postures play a direct role in creative cognition[27–29], potentially by enhancing the clarity of mental imagery—a mechanism proposed by Matheson and Kenett[30] to explain how physical actions liberate creative thinking (see also[31]). Neuroimaging evidence further corroborates this sensorimotor-creativity link: creative tasks consistently recruit functional coupling within sensorimotor brain regions[30,32–34], suggesting that mental imagery operates as an embodied cognitive strategy[14,29,31]. Second, explicit investigations into mental imagery reveal its multidimensional contributions to creativity. Researchers have examined the link between spatial visualization ability, self-reported visual imagery and creative performances[16,35], respectively, and using imagery-based tasks to generate creative products[36,37]. However, as Kozhevnikov et al.[38] argued, such approaches often oversimplify both constructs, that creativity and mental imagery are not single homogeneous constructs but vary across domains. For instance, artistic versus scientific creativity differentially engage object versus spatial visualization abilities[38,39]. Thus, examining mental imagery in domain-specific creativity is essential.

The cognitive mechanisms of mental imagery supporting creative cognition likely operates through semantic memory, given its integral roles in prospection, mental imagery and divergent thinking[13]. Semantic memory stores facts, concepts, and general knowledge, which can be retrieved and combined in novel ways to foster creativity[40]. Traditionally, mental imagery has been associated with episodic or visual memories[41–45]. For instance, neuroimaging studies indicate that the brain regions involved in mental imagery overlap with those supporting episodic memory retrieval[46], which is logical since mental imagery often involves concrete, visualized memory aspects.

However, research also indicates that mental imagery underpins semantic memory[47–49]. Mental imagery serves as modality-specific inputs to semantic representation and then integrates with semantic control processing[50]. Firstly, mental imagery contributes to semantic representation. The hub-and-spoke model posits that cross-modal interactions are mediated by a bilateral anterior temporal lobe (ATL) hub, which itself contains no semantic features but represents the semantic similarity among concepts[50–52]. Multimodal imagery, which refers to more than one sensory modality, is thus recruited and interacts with a cross-modal hub to form generalizable concepts[50,53]. Similarly, the neuro-cognitive model of semantic memory proposed by Fernandino et al.[54] supports the notion of hierarchical sensory-motor integration as an essential architectural feature of semantic memory. Additionally, empirical research underscores the utility of embodied strategies, such as action or body simulations, in verbal divergent thinking tasks[14,29,31]. Secondly, neuroimaging evidence further revealed that imagery engagement in semantic cognition is contextually optimized. Studies found that when the linguistic context emphasizes action properties, greater activation is observed in brain regions relevant for coding action information[55]. Recent studies support the finding by revealing that visual feature matching tasks require a visually focused brain state, engaging control regions for goal maintenance and prioritization of relevant visual knowledge, which must be functionally distinct from heteromodal conceptual hubs (i.e., higher-order association cortices) yet tightly integrated with task-specific sensory areas[56]. These findings align with the controlled semantic cognition framework, which proposes that top-down processes modulate activation within semantic representational systems to support goal-directed behavior[50]. Therefore, mental imagery participates in semantic memory processing and is manipulated by top-down control processes.

Semantic memory is particularly important for creativity[7,57–59]. According to the associative theory of creativity[60,61], semantic memory structure reflects individual differences in creativity. Highly creative individuals possess a relatively flat associative hierarchy (numerous and weakly related associations to a given concept; e.g.,[62,63]), allowing them to easily retrieve and combine remote associative elements. Modern computational network science tools have been applied to examine the structure and search process that operates over semantic memory related to creativity[58,59] by conceptualizing semantic memory structure as a network comprising nodes (semantic memory or concepts) and edges (semantic similarity)[33,58,59,64–67]. Abundant studies have investigated the role of semantic memory structure in creative abilities, by conducting experiments involving remote association, divergent thinking tasks and semantic judgment rating tasks at the group or individual level[33,66–74]. These studies highlight that higher creative individuals exhibit a more flexible semantic memory structure featured with shorter distances between concepts, fewer subcommunities in their networks and flexible and progressive retrieval patterns, compared to lower creative individuals[58]. Moreover, researchers have applied computational methods to model the search process of semantic memory, and found that higher creative individuals visit farther, more unique, and weaker nodes in their semantic memory networks[63,73,75–79]. Therefore, the impact of mental imagery on creative cognition may be mediated by its influence on semantic memory.

Therefore, we focused on two top-down mechanisms within semantic memory through which mental imagery may facilitate creativity. The first is semantic integration, which enables the binding of multimodal features into coherent conceptual representations. Creative cognition requires the ability to search more broadly in memory and connect weakly related concepts and thus generate creative ideas[58], a capacity supported by semantic integration. The second is semantic reorganization, which reflects the flexible restructuring of conceptual relationships to meet task demands. Creative cognition involves controlled semantic search, which benefits from cognitive flexibility to facilitate spread of activation in semantic memory[67,71]. To operationalize these mechanisms, we use two semantic indicators derived from creative writing: (1) *semantic integration*, reflecting the ability to connect distant associative concepts and weave rich ideas into cohesive narratives[80], and (2) *semantic network robustness*, capturing the flexible reorganization of concepts, directly measuring cognitive flexibility[71,81,82]. Semantic integration primarily reflects the breadth of associative thinking in connecting distant concepts to form novel wholes. In contrast, semantic network robustness highlights the cognitive flexibility in adaptively restructuring conceptual relationships. This framework is grounded in the roles of mental imagery in semantic processing, as well as the established role of semantic memory in creativity. Notably, previous studies treated semantic features as stable traits[67,69,78], but quantifying these features is challenging due to lengthy

experiments and methodological considerations[68,83,84]. Instead, we view them as dynamic state variables reflecting dynamic cognitive processes[82,85].

To investigate how mental imagery supports creative cognition, we employed two approaches. First, we examined correlations between self-reported imagery vividness (via the Plymouth Sensory Imagery Questionnaire, Psi-Q)[86] and creative writing performance[23,25,87,88]. Second, participants completed creative writing tasks using either imagery or semantic understanding strategies, inspired by Paivio's dual coding theory[89], which suggested two encoding modes in human cognition: verbal and non-verbal (imagery) coding. Therefore, a feasible strategy is to have individuals represent stimuli and encoding information in imagery or verbal forms and compare creative performance. For example, studies have compared the induction of anxiety and positive emotions when reading texts under imagery and verbal conditions[90,91]. In the imagery condition, participants generated multi-sensory imagery based on text descriptions, while in the verbal condition, they focused on the text's meaning. In this study, we adopted a similar paradigm, asking participants to complete creative writing tasks using either imagery or semantic understanding strategies.

Neuroscience studies have provided indirect support for the role of mental imagery in creative cognition through findings related to sensorimotor regions. For instance, research on highly creative individuals, such as scientists and artists, reveals heightened activation in brain areas associated with motor imagery, suggesting that mental imagery may drive their creative processes[92]. Similarly, Einstein's enlarged parietal regions, linked to his reliance on sensory-based thinking (e.g., visual imagery), may have underpinned his exceptional cognitive abilities[93]. Further evidence comes from visual creativity studies, which show that enhanced activation of the visual cortex during creative tasks may reflect not only visual information processing but also greater engagement of visual imagery processes. Specifically, the left fusiform gyrus, closely tied to visual imagery, plays a critical role in mental rotation and semantic representation retrieval during tasks like visual design[94]. A recent heterarchical model of visual mental imagery proposed that the left fusiform gyrus connects high-level visual areas processing domain-specific information with other regions in the temporal lobe implicated in memory- and semantic-related processes[95]. Additionally, studies on uncommon tool use demonstrate flexible activation within the "tool network", linking action and visual information in dorsal stream areas[96]. Network neuroscience methodologies further highlight the significance of neural integration within sensorimotor regions for creativity[32,97]. However, direct evidence of the neural mechanisms underlying mental imagery's involvement in creative cognition remains limited.

We aim to investigate the brain reconfiguration patterns of creative cognition during imagery encoding. Based on creativity neuroscience research, extensive neuroimaging studies have suggested that creative cognition involves the integration of multiple cognitive processes, expanding from unimodal to multimodal brain regions[9,98,99]. The interaction between the executive control network and the default mode network constitutes the fundamental architecture of creative cognition, elucidating how self-generated thought and cognitive control collaboratively drive its mechanisms[10,11,100–104]. Unimodal networks contribute to creative cognition as well, such as somatomotor and visual networks[32,33]. Research on brain functional connectivity (FC) has advanced our understanding of the specific functions of brain regions involved in creative cognition. However, how do these regions integrate into large-scale networks? Community detection approaches identify densely interconnected node modules, revealing how network organization supports cognitive functions[105]. Previous node-centric functional brain community research suggests functional brain organization patterns vary across individuals, over time, and during different task performances[106–108]. Studies have also revealed the organizational patterns based on node FC related to creativity[34,99,109,110].

There is evidence uncovering that brain organizations of creative cognition may be much more complex distributed. Dynamic reconfiguration-based methods have revealed that heteromodal networks reconfigure their FCs and communities in various time windows during creative thinking[11,34,99,111,112], suggesting a single network can have varying

network interactions during creative cognition. Further studies found that heteromodal networks collaborate but also demonstrate distinct FC patterns to support creative thinking and achievement[9,113]. For example, Beaty et al.[109] found distinct connectivity dynamics for subnetworks within the fronto-parietal control network (FPCN) with the default mode network. Specifically, FPCNa (including rostrolateral prefrontal cortex, superior frontal gyrus, and anterior intraparietal sulcus) showed the greatest coassignment to the default network, reflecting dynamic reconfiguration during creative cognition. The complex roles of the FPCN in creative cognition are further revealed in a recent study[114], which investigated the separate neural foundations of two creativity components, showing that the FPCN has less weight in both the novelty and appropriateness neural contributions. These findings imply that functional brain organizations are highly complex, extending beyond a simple model of strictly segregated and non-overlapping brain modules[115,116], with a brain region playing multiple functional roles in creative cognition. However, the overlapping brain organization patterns of creative cognition, particularly during imagery encoding, have yet to be fully explored.

Recently, an edge-centric framework was proposed to identify overlapping functional communities in brain networks, leveraging edge functional connectivity (eFC) to reveal previously obscured overlapping modular organization[117]. This recent method represents pairwise functional interactions between network edges[118,119]. It involves computing the strength of functional connections between node pairs by unwrapping Pearson correlations over time, generating brain activity signals for each edge and observing fluctuations in their weights. Edge time series estimate edge-related structures, known as eFC, which measures the similarity of time-varying co-fluctuations across the brain[115]. High-amplitude eFC indicates strong similarity in communication patterns between edges, while low-amplitude eFC suggests independent fluctuations[120]. Clustering eFC classifies edges into non-overlapping edge communities. However, when these edge communities are mapped back to nodes, the partitions overlap, meaning a single node can be part of multiple edge communities[115,121]. eFC extends and complements traditional node-centric brain network models, which measure inter-regional communication through temporal correlations. That is, strong functional connections are thought to reflect the time-averaged strength of communication between brain regions. eFC tracks how communication patterns evolve over time and ultimately assesses whether similar patterns are occurring in the brain simultaneously[120]. This approach provides a fine-grained perspective on brain network organization, capturing co-fluctuations in neural activity that may be critical for understanding complex cognitive processes.

Here, we utilized the edge-centric functional networks approach to analyze the synchrony of brain communication patterns, capturing dynamic brain organization during creative cognition under mental imagery and semantic understanding conditions. Recent studies found that edge-centric functional brain community distributions provide new insights into human cognition. A study found that edge communities vary across different brain system, such as heteromodal systems are more diverse than sensory systems and are more likely to form their own clusters[115]. A recent study found brief periods of high-amplitude brain network connectivity in mice and humans, suggesting that these dynamic patterns are consistent across species[122]. Another study identified a general bridging factor in depression-anxiety comorbidity using edge-centric connectomes, revealing its robust generalizability[123]. The edge-centric framework is particularly well-suited for studying creative cognition due to its ability to uncover overlapping functional organization. Creative cognition involves the dynamic integration of multiple cognitive processes supported by interactions between large-scale brain networks like the default mode network and executive control network[97,101,102]. eFC captures the moment-to-moment fluctuations in communication patterns, enabling the identification of overlapping functional communities that reflect the flexible reconfiguration of brain networks during creative tasks.

To sum up, we hypothesize that mental imagery enhances creative cognition by facilitating the integration and reorganization of concepts in

semantic memory, alongside investigating the overlapping brain functional organization patterns of creative cognition involving imagery encoding. To test this, we first conducted two behavioral studies to examine the relationship between mental imagery and creative writing performance, focusing on the functional roles of semantic memory. Additionally, we conducted an fMRI study to explore the neural mechanisms underlying creative writing tasks involving imagery encoding. Our findings offer a deeper understanding of how mental imagery supports creative cognition, from both cognitive and neural perspectives, particularly through semantic memory processes and edge-centric brain functional network dynamics.

## Results
### Mental imagery, semantic features and creative writing
In Study 1, Spearman's correlation analysis was conducted on the dimensions of Psi-Q, creative writing scores (based on the three-words creative

**Table 1 | Relationship between creativity and mental imagery and semantic features**

| | CW_subjective | | CW_objective | |
|---|---|---|---|---|
| | $r_s$ | p | $r_s$ | p |
| Psi-Q average | 0.120 | 0.240 | 0.150 | 0.141 |
| Psi-Q vision | 0.168 | 0.098 | 0.085 | 0.403 |
| Psi-Q sound | 0.082 | 0.424 | 0.055 | 0.589 |
| Psi-Q smell | 0.121 | 0.236 | 0.103 | 0.311 |
| Psi-Q taste | 0.121 | 0.235 | 0.096 | 0.348 |
| Psi-Q touch | **0.278** | **0.006**** | **0.305** | **0.002**** |
| Psi-Q bodily sensation | 0.008 | 0.935 | 0.107 | 0.295 |
| Psi-Q emotional feeling | −0.045 | 0.657 | 0.066 | 0.520 |
| SI | **0.373** | **<0.001***** | **0.337** | **<0.001***** |
| SNR | **0.550** | **<0.001***** | **0.380** | **<0.001***** |
| SNR_noise | **0.550** | **<0.001***** | **0.381** | **<0.001***** |
| SNR_shuffling | **0.548** | **<0.001***** | **0.388** | **<0.001***** |

*CW* subjective and objective ratings of creative writing, *SI* semantic integration, *SNR* semantic network robustness, *SNR_noise* semantic network robustness added noise, *SNR_shuffling* semantic network robustness after shuffling.
Boldface denotes statistical significance ($p < 0.05$) after false discovery rate correction for multiple comparisons. **$p < 0.01$, ***$p < 0.001$.

writing task), and semantic feature indicators to explore the relationship between creative writing, mental imagery and semantic features (Table 1). After multiple comparison correction[124], it was found that both subjective and objective scores of creative writing performance were significantly positively correlated with Psi-Q touch score, as well as with semantic integration and semantic network robustness. However, other Psi-Q dimensions did not significantly correlate with creative writing scores. Semantic network robustness correlated with creative writing scores after adding noise and shuffling links, showing that it was a stable percolation process and the correlations between robustness and creative writing scores were mainly determined by the structural characteristics of the network, rather than edge weights. Furthermore, there was a significant positive correlation between the subjective and objective scores of creative writing performance (Supplementary Fig. 1). The complete correlation analysis results among all variables (both uncontrolled and age/gender-adjusted) are shown in Supplementary Fig. 2.

Mediation analyses were conducted to explore the semantic memory processing that may be involved, with two semantic feature indicators serving as the mediator in the model, respectively. With the touch score of Psi-Q as the independent variable and the creative writing scores (both subjective and objective) as the dependent variable, the mediation models with both semantic feature indicators were established (Fig. 1). Results of mediation models controlling for age and gender are represented in Supplementary Table 1.

For semantic integration, in the *Subjective creative writing score model* we found that the indirect effect was significant ($\beta = 0.096$, $p = 0.040$), with 95% CI = [0.0037, 0.2430], the direct effect was significant ($\beta = 0.217$, $p = 0.015$), the total effect was significant ($\beta = 0.313$, $p < 0.001$) (Fig. 1A). In the *Objective creative writing score model*, we found that the indirect effect was significant ($\beta = 0.084$, $p = 0.031$), with 95% CI = [0.0003, 0.0048], the direct effect was significant ($\beta = 0.218$, $p = 0.019$), the total effect was significant ($\beta = 0.302$, $p < 0.001$) (Fig. 1B).

For semantic network robustness, in the *Subjective creative writing score model*, we found that the indirect effect was significant ($\beta = 0.171$, $p = 0.007$), with 95% CI = [0.0530, 0.3708], the direct effect was not significant ($\beta = 0.142$, $p = 0.129$), the total effect was significant ($\beta = 0.313$, $p < 0.001$) (Fig. 1C). In the *Objective creative writing score model*, we found that the indirect effect was significant ($\beta = 0.105$, $p = 0.018$), with 95% CI = [0.0005, 0.0059], the direct effect was significant ($\beta = 0.197$, $p = 0.042$); the total effect was significant ($\beta = 0.302$, $p < 0.001$) (Fig. 1D).

The results indicate that mental imagery promotes creative writing performance via semantic integration and semantic network robustness,

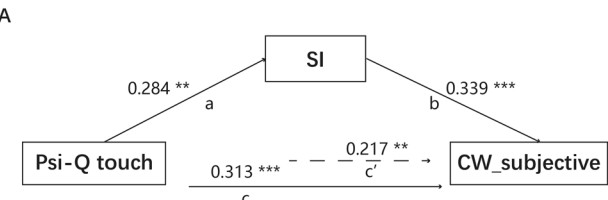

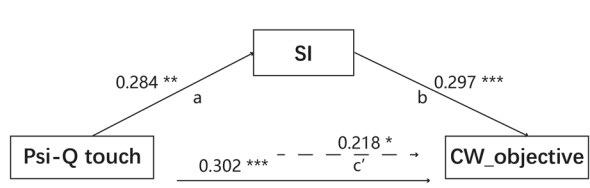

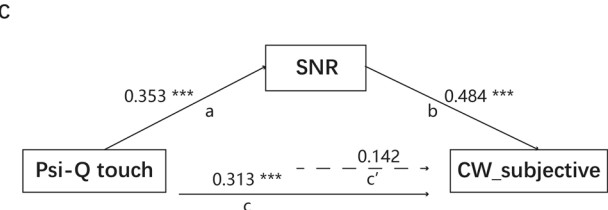

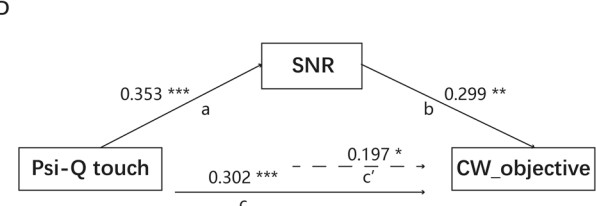

**Fig. 1 | Mediation analyses. A–D** Psi-Q touch serves as the independent variable, while creative writing performance (both subjective and objective score) serves as the dependent variable, with two semantic feature indicators serving as the mediator in the mediation model, respectively. SI semantic integration, SNR semantic network robustness. * - $p < 0.05$, ** - $p < 0.01$, *** - $p < 0.001$.

and shows consistency between subjective and objective scores of creative writing performance.

## Imagery strategy enhances creative writing performance

In Study 2, by comparing the creative writing performances under two encoding modes, i.e., non-verbal (imagery) and verbal coding, we examine whether mental imagery strategies could enhance creative writing performance. Self-ratings of participants in instructional phase and across ten trials were analyzed on an 1–11 scale. Self-rated scores during the instructional phase averaged $9.01 \pm 1.43$ for mental imagery vividness and $9.76 \pm 1.24$ for semantic understanding clarity in lemon-cutting, and $9.34 \pm 1.51$ and $9.38 \pm 1.38$ for comprehension of mental imagery and semantic understanding concepts, respectively. One-sample t-test analyses confirmed these scores were significantly above the midpoint of 6 (all $p$'s $< .001$), indicating participants clearly understood the task requirements (Fig. 2A).

Averages of self-rated scores across 10 trials per condition revealed ranges of $8.68 \pm 1.25$ for mental imagery vividness and $8.36 \pm 1.50$ for its strategy usage; $9.06 \pm 1.49$ for semantic understanding clarity and $8.32 \pm 1.55$ for its strategy usage. One-sample t-test analyses revealed that mental imagery vividness and usage, as well as semantic understanding clarity and usage, were significantly above the average score of 6 (all $p$'s $< 0.001$) (Fig. 2A). These findings support the validity of conditions and employing the strategies in creative writing tasks.

Next, objective rating of creative writing and semantic features were compared between the two conditions (Table 2). To quantify the creativity of creative writing performance, only the objective rating was used here, as the correlation coefficient between the objective rating of creative writing and the Psi-Q touch score was higher than that of the subjective rating in Study 1. Paired-samples $t$-test analyses indicated that the objective creative writing scores under mental imagery condition were significantly higher than those under semantic understanding condition, $t(67) = 4.967$, $p < 0.001$, $d = 0.603$ (Fig. 2E), and no significant difference in semantic integration between two conditions, $t(67) = -0.205$, $p = 0.839$, $d = -0.025$. Semantic network robustness under mental imagery condition was significantly higher than that under semantic understanding condition, $t(67) = 2.514$, $p = 0.014$, $d = 0.305$ (Fig. 2F). Figure 2B illustrates the percolation process for a participant under two different conditions. Figures 2C, D, respectively, depict the semantic networks under these conditions, with edge weights thresholded at a Pearson correlation coefficient of 0.5.

To examine the significance of our findings, we applied noise analysis and link shuffling analysis (Table 2). Semantic network robustness with added noise was significantly higher in the mental imagery condition than in the semantic understanding condition, $t(67) = 2.512$, $p = 0.014$, d = 0.305. The result suggests both conditions had a stable percolation process with significant different semantic network robustness. We then compared semantic network robustness after shuffling links in the network, and no significant difference was found between the two conditions, $t(67) = 0.852$, $p = 0.397$, $d = 0.103$. This finding suggests that the difference between the original semantic network robustness of the two conditions was driven by the structure of the network and not by the edge weights.

These results suggest that mental imagery strategies enhance creative writing performance and are reflected in the flexible reorganization of semantic memory.

## Brain edge community patterns involving imagery encoding

The analysis of edge community overlap during creative writing under mental imagery and semantic understanding conditions revealed that multiple edge communities were commonly present within a brain network, indicating parallel pathways for information communication within a brain network (Fig. 3A, B). The Kruskal-Wallis test indicated significant differences in the median values of community overlap (measured by normalized entropy) across nodes in the 17 brain networks under mental imagery condition, $H(16) = 245$, $p < 0.001$, $\varepsilon^2 = 0.613$, and semantic understanding condition, $H(16) = 239$, $p < 0.001$, $\varepsilon^2 = 0.599$. Post hoc Dwass-Steel-

Critchlow-Fligner test found that nodes in the $SMN_B$, $DAN_A$, $DAN_B$, $SAL_A$, $SAL_B$, and $LIM_A$ had the highest community overlap (with no significant differences between the medians of these 6 networks), and nodes in the $LIM_B$ networks had the lowest community overlap under mental imagery condition (Fig. 3A, B). Nodes in the $SMN_A$, $SMN_B$, $DAN_A$, $DAN_B$, $SAL_A$, $SAL_B$, and $LIM_A$ had the highest community overlap (with no significant differences between the medians of these 7 networks), and nodes in the $LIM_B$ networks had the lowest community overlap under semantic understanding condition (Fig. 4A, B). Violin plots of edge community overlap from $k = 2$ to 20 under two conditions are presented in Supplementary Figs. 3 and 4. Results of the Dwass-Steel-Critchlow-Fligner test are presented in Supplementary Data 1.

Then, Study 3 investigated whether the distribution of edge communities within the brain network is uniform or varies. The Kruskal-Wallis test indicated significant differences in the median values of edge community similarity across nodes in the 17 brain networks under mental imagery condition, $H(16) = 240$, $p < 0.001$, $\varepsilon^2 = 0.603$, and semantic understanding condition, $H(16) = 234$, $p < 0.001$, $\varepsilon^2 = 0.588$, revealing that the distribution of edge communities varied within different brain networks. Post hoc Dwass-Steel-Critchlow-Fligner test found that nodes in the $SMN_B$, $DAN_A$ had the highest community similarity (with no significant differences between the medians of these 2 networks), and nodes in $SMN_A$, $SAL_A$, $FPCN_A$, $FPCN_B$, $DMN_A$, $DMN_B$, and $DMN_C$ networks had the lowest community similarity (with no significant differences between the medians of these 7 networks) under mental imagery condition (Fig. 3C, D). Nodes in the $SMN_B$, $DAN_A$, $LIM_A$ and TP had the highest community similarity (with no significant differences between the medians of these 4 networks), and nodes in the $SAL_A$, $FPCN_A$, $FPCN_B$, $FPCN_C$, $DMN_A$ and $DMN_C$ networks had the lowest community similarity (with no significant differences between the medians of these 6 networks) under semantic understanding condition (Fig. 4C, D). Violin plots of edge community similarity from $k = 2$ to 20 under two conditions are presented in Supplementary Figs. 5 and 6. Results of the Dwass-Steel-Critchlow-Fligner test are presented in Supplementary Data 1.

## Discussion

Mental imagery plays a functional role in human cognition[92,125], yet its specific contribution to creativity remains largely unexplored. In the current study, we investigated how mental imagery supports creative cognition through three studies. We first conducted two behavioral studies to explore the cognitive mechanisms by which mental imagery affects creative cognition, particularly from the perspective of semantic memory. This was followed by an fMRI study that examined the distribution characteristics of edge communities in brain activity involving imagery encoding in creative cognition.

### The role of mental imagery in creative cognition

To test how mental imagery facilitates creative cognition, we first investigated the correlation between them. Study 1 revealed that the more vivid touch imagery, the better the creative writing performance. Mediation analyses then revealed that touch imagery vividness positively influenced creative writing performance through two distinct semantic features: semantic integration and semantic network robustness. In Study 2, we further examined how mental imagery supports creative cognition by comparing creative writing performance under two conditions: mental imagery and semantic understanding. The results revealed that creative writing performance was significantly better in the mental imagery condition. Additionally, semantic network robustness was higher in the mental imagery condition, while there was no significant difference in semantic integration between the two conditions.

The correlation analysis in study 1 highlighted that, among various sensory modalities, touch is particularly significant for creative cognition, likely due to its role as a proximal sense[126,127] that allows for the direct perception of the physical presence of objects. Taylor et al.[128] posited that what we touch is perceived as more real than what we see. Touch is also a

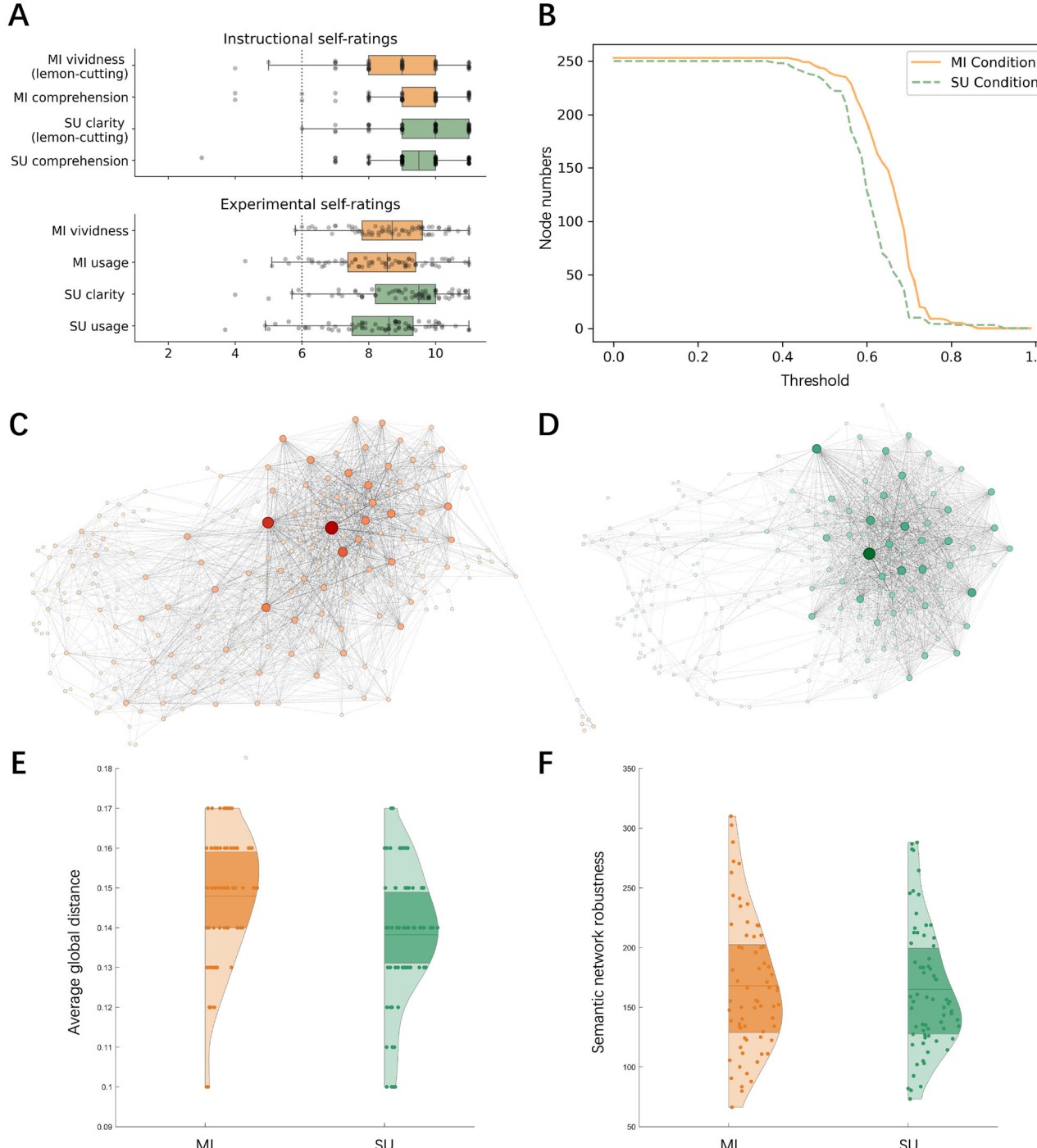

**Fig. 2 | Creative writing performance under two conditions. A** Self-rated scores of participants in instructional phase and across ten trials (averaged). Dashed line represents the median of the scores. Five horizontal lines in the boxplots indicate key percentiles (from left to right: minimum, 25th percentile, median, 75th percentile, and maximum. **B** Percolation process for a participant under two conditions. The x-axis represents the weight threshold. **C, D** Semantic networks under mental imagery and semantic understanding conditions, respectively, for the same participant as in (**B**). Edge weights are thresholded at Pearson correlation coefficient = 0.5

for visualization purposes. Node size and color are proportional to centrality computed from unthresholded networks. **E** The violin plot of objective ratings of creative writing performance under two conditions. The y-axis refers to the average of global distance for ten stories under the same condition. Five horizontal lines indicate key percentiles (from bottom to top: minimum, 25th percentile, median, 75th percentile, and maximum. **F** The violin plot of semantic network robustness under two conditions.

crucial component of social interaction and relationship formation, as expressed in poetry—"Hapticality, the capacity to feel through others, for others to feel through you, for you to feel them feeling you" emphasizing the interactivity and emotional exchange of touch[129]. Previous research has linked touch with higher cognitive processes. For example, Galton[130] posited

that the sharper one's sensory discrimination is, the broader the cognitive scope. Li et al.[131] found that touch measures (touch discrimination ability, touch pressure sensitivity) accounted for 20.8% of the total variance in intelligence. Additionally, active touch has been shown to positively influence creative performance, particularly in product ideation[132]. While

previous studies have not fully elucidated how touch affects cognition, it can be hypothesized that touch provides a more intimate and direct way of perceiving the world compared to other senses. Touch is central to many perceptual activities, forming a distinct yet overlapping touch memory that influences cognition[133].

Our results further support the hypothesis that mental imagery contributes to creative cognition by facilitating the processing of semantic memory through semantic integration and reorganization. Specifically, vivid touch imagery enhances creative writing performance by promoting the integration of novel concepts and enabling their flexible manipulation within semantic memory. These findings align with previous hypotheses

### Table 2 | Objective rating of creative writing and semantic features comparisons between two conditions

|  | MI | SU | MI vs. SU |
|---|---|---|---|
| CW_objective | 0.148 (.02) | 0.138 (.02) | $t(67) = 4.967^{***}$ |
| SI | 0.47 (.01) | 0.47 (.01) | $t(67) = -0.205$ |
| SNR | 168.08 (56.40) | 165.03 (53.13) | $t(67) = 2.514^{*}$ |
| SNR_noise | 168.09 (56.40) | 165.03 (53.13) | $t(67) = 2.512^{*}$ |
| SNR_shuffling | 183.33 (60.90) | 182.32 (58.35) | $t(67) = 0.852$ |

MI mental imagery, SU semantic understanding, CW_objective objective rating of creative writing, SI semantic integration, SNR semantic network robustness, SNR_noise semantic network robustness added noise, SNR_shuffling semantic network robustness after shuffling. $^{*}p < 0.05$, $^{***}p < 0.001$.

that semantic memory integrates sensory input into higher cognitive processes. For instance, embodied semantics theories propose that conceptual knowledge is grounded in sensory-perceptual experiences[134,135]. In the information processing model of imagination proposed by Abraham and Bubic[13], semantic memory plays a foundational role by abstracting content from specific sensory, motor, and affective experiences and engaging in broader semantic operations, such as creative thinking. Our results reveal that mental imagery contributes to creative cognition by facilitating semantic memory processing in two ways: by establishing connections to diverse semantic memory elements, and by flexibly manipulating connections between these elements.

The contributions of mental imagery to creative cognition can be further understood through its distinct cognitive facets. Grounded in the theoretical models proposed by Kosslyn and colleagues[125,136], visual mental imagery can be divided into two principal facets: *Representation*, involving the generation and maintenance of imagery, and *Manipulation*, involving the examination and alteration of imagery. Emerging evidence suggests these facets differentially scaffold creativity. The representation of imagery provides raw material for creative ideation. For example, Kaufman and Kaufman[23] reported that blocked writers are more likely to report low levels of positive, constructive mental imagery, and a lower level of vividness in their current work-related mental imagery activity. Another study found that young children who engaged in visual-auditory mental imagery demonstrated greater musical compositional creativity than those in the control group[37]. These findings indicate that mental imagery representation is related to creative performance. The manipulation of imagery enables restructuring of mental content. Pearson and Logie[137] discovered that external manipulation of mental imagery facilitated reference frame

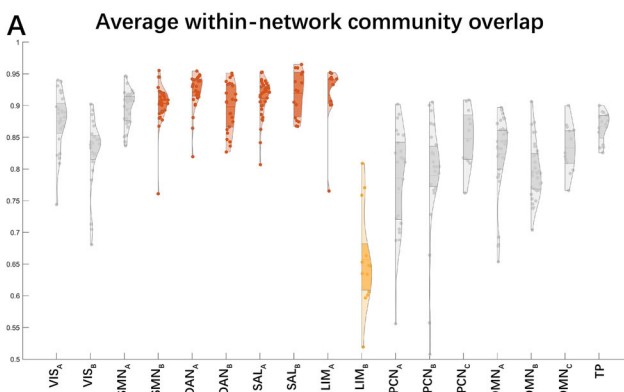

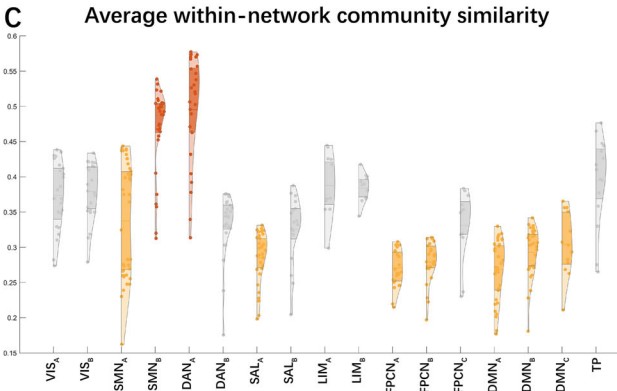

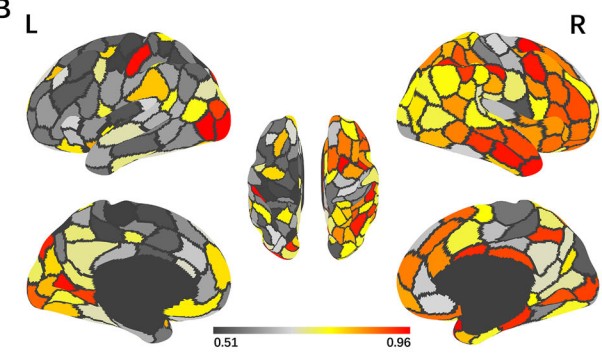

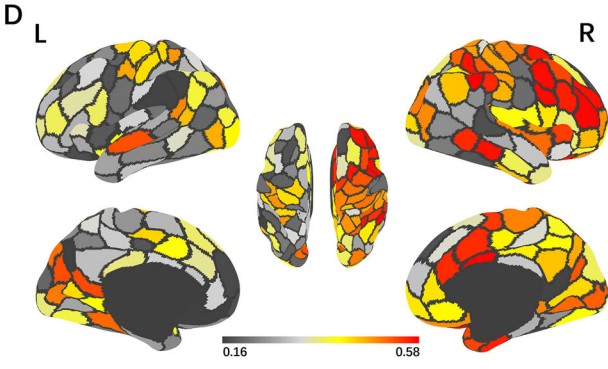

**Fig. 3 | Average edge community overlap and edge community similarity within the brain network under mental imagery condition. A, C** Violin plots of average edge community overlap and edge community similarity for each node (i.e., brain region) within the Yeo-Krienen 17 networks, respectively. Orange violins indicate that nodes within the network have the highest normalized entropy or highest similarity in their community assignments, while yellow violins indicate the opposite trends. **B, D** Depict cortical surface projections of average edge community overlap and edge community similarity, respectively. VIS visual network, SMN sensorimotor network, DAN dorsal attention network, SAL salience network, LIM limbic network, FPCN frontoparietal control network, DMN default mode network, TP temporal-parietal network.

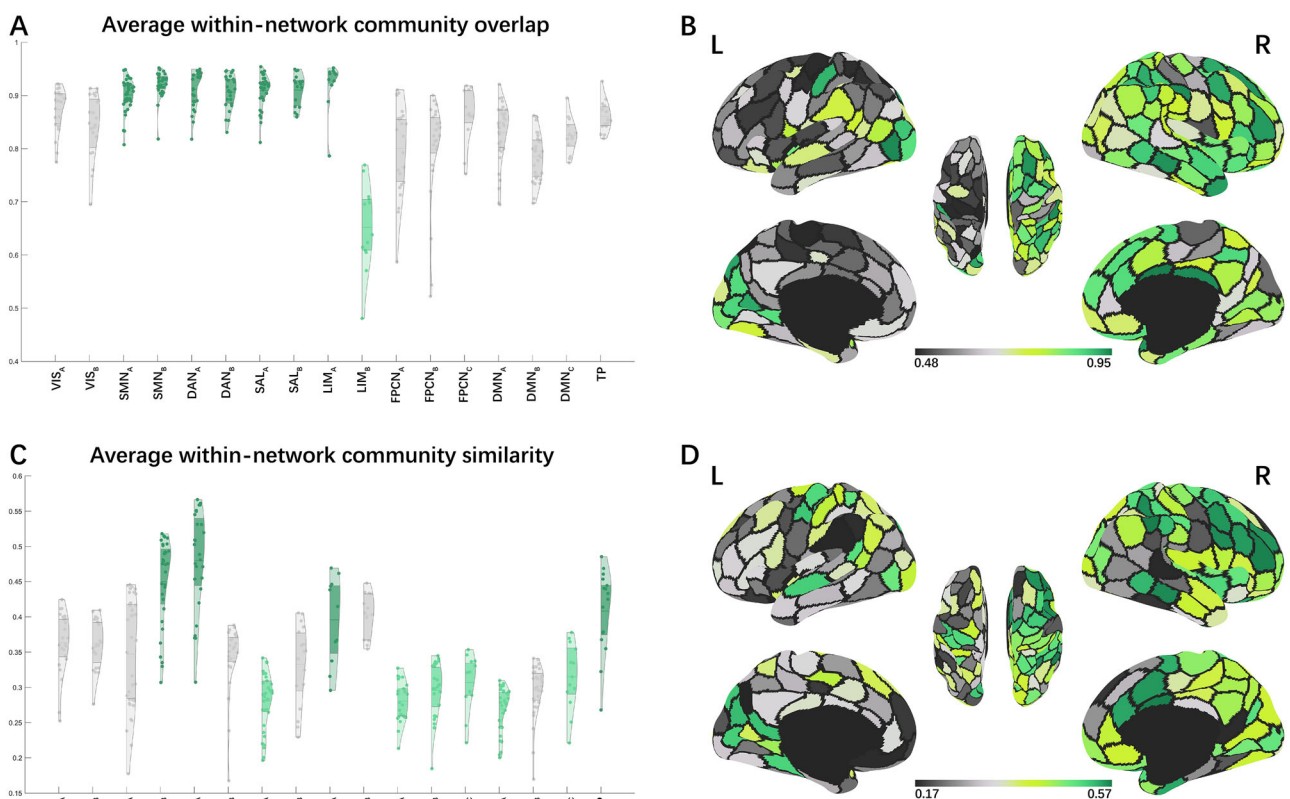

**Fig. 4 | Average edge community overlap and edge community similarity within the brain network under semantic understanding condition. A–D** Description is similar to Fig. 3 legend. VIS visual network, SMN sensorimotor network, DAN dorsal attention network, SAL salience network, LIM limbic network, FPCN frontoparietal control network, DMN default mode network, TP temporal-parietal network.

selection and change during creative integration. Palmiero et al.[16,35] found that visual creativity was positively correlated with image transformation imagery ability. Additionally, a study found that imagery manipulation training improved undergraduate dance students' domain-general flexible thinking performance[36]. Therefore, mental imagery also supports creative cognition through its manipulation facet. However, the current study did not differentiate between imagery representation and manipulation, and future research should investigate their distinct roles in creative cognition mediated by semantic cognition.

We further found that creative writing performance under mental imagery condition was superior to that under semantic understanding condition. This result could be attributed to the distinct ways in which mental imagery and semantic understanding strategies influence semantic memory processing, thereby differentially impacting creative writing performance. Semantic understanding strategies may rely on top-down processing, where controlled and focused retrieval of well-defined concepts occurs. This focused retrieval can streamline cognitive resources but may also result in functional fixedness by narrowing the scope of available concepts and limiting the potential for novel associations[138]. In contrast, mental imagery strategies typically engage bottom-up processing, which is more exploratory and involves ambiguous targets[139]. This disassembling approach may encourage the activation of a broader range of semantic associations, helping to avoid functional fixedness and thus fostering creativity.

Our findings regarding the distinct patterns of semantic integration and semantic network robustness under the two conditions further indicate that these strategies differentially influence semantic memory processing. While both strategies lead to rich semantic integration, flexible reorganization of concepts is significantly more pronounced under mental imagery condition. This suggests that semantic reorganization, beyond the role of semantic integration, plays a critical role in how mental imagery enhances creativity.

In creative activities, mental imagery may play a more substantial role through semantic reorganization compared to integration. For instance, a study categorized accomplished artists and scientists during their peak productivity into high and medium creativity groups[92]. During a verbal creative task, no significant difference in activation of the primary visual cortex was found between the groups. However, significant differences were observed in areas related to motor imagery (as participants imagined their actions), with the high creativity group more vividly imagining manipulation and transformation of visual materials. The lack of significant difference in primary visual cortex activation may indicate that both medium and high creators visualized images, while the significant difference in motor imagery-related areas may imply that mental imagery manipulation allowed further symbolization and processing of images, promoting the development and connectivity of neural pathways and forming novel, multimodal, and condensed creative representations. Another study found that cue set-size (reflecting the semantic richness of associated cues) differentially impacted the quality and quantity of generated creative uses, with low-association, sparse alternate uses task (AUT) cues favoring originality over fluency compared to high-association, rich cues, indicating that cognitive control processes could enhance idea generation when conceptual knowledge was limited[62]. This suggested that creative cognition requires overcome of knowledge constraints and conceptions expansion[140]. The results of Study 2 indicates that while imagery provides a rich internal resource for creative cognition, the creative process should surpass mere integration, involving complex reorganization of cognitive elements[9,92].

## Neural basis underlying creative cognition involving imagery encoding

By constructing brain edge communities and comparing the distribution of edge communities within traditional brain networks, we explored how brain networks participated in creative cognition under both conditions. The

formation of overlapping brain edge communities suggests that the brain's intrinsic functions dynamically combined. Moreover, the involvement of a node in multiple states or functions of information communication implies that the brain's information exchange is flexible and complex due to this multiple community affiliation.

We found the highest community overlap and similarity within $SMN_B$ under both conditions, and the lowest community similarity within $SMN_A$ under mental imagery condition. This suggests the $SMN_B$ participated in uniform and homogeneous communication under both conditions, whereas the $SMN_A$ participated in variable communications patterns under mental imagery condition. The $SMN_B$ mainly includes ventral parts of anterior central gyrus (PreCG) and posterior central gyrus (ProCG), terminating in the auditory cortex. The $SMN_A$ consists of the dorsal and medial parts of PreCG and ProCG. According to the work of Kong et al.[141], the $SMN_B$ is involved in auditory processing and face somatomotor activity, whereas the $SMN_A$ is associated with foot and hand somatomotor functions. The communication patterns within the $SMN_B$ may reflect its specialized role in auditory input processing or weak auditory imagery[142]. The $SMN_A$ participated in various brain communication patterns likely reflects sensorimotor simulations in creative cognition. For a review, see Matheson and Kenett[30]. This supports the view that mental imagery operates as a dynamic information processing mechanism (rather than a static structural feature), driven by working memory's capacity to encode, maintain and manipulate sensory representations[143].

The highest community overlap within the DAN and SAL networks were found, along with the highest edge community similarity in $DAN_A$ and the lowest in $SAL_A$ under both conditions. Functionally, DAN is known to maintain goal-directed attention, whereas SAL detects behaviorally salient events and reorients attention[144–147]. The high overlap in DAN and SAL likely reflects their capacity to integrate goal-relevant signals and salient stimuli, as demonstrated in visual search[148] and divergent thinking tasks[79], a critical process for creative thinking that requires both sustained focus on internal ideas and flexible capture of task-related cues from environment[34].

The highest edge community similarity in $DAN_A$ suggest that it engages in highly coordinated information processing during the Thinking phase under both conditions. This aligns with prior findings that DAN reduces its global connectivity (degree centrality) during goal-directed tasks[149], possibly reflecting a shift from broad network integration to focal within-network synchronization. In our study, $DAN_A$'s high edge community similarity may reflect it is potentially supporting stable attentional engagement. We found $SAL_A$ showed the lowest edge community similarity. SAL is known to mediate switches between DMN and FPCN, suppressing task-irrelevant self-generated thoughts to guide task-relevant behavior[150]. $SAL_A$'s heterogeneous community distribution may reflect cooperation between brain networks associated with spontaneous thought and cognitive control[112].

Sequentially, $LIM_A$ showed the highest edge community overlap under both conditions, and exhibited the highest edge community similarity under semantic understanding condition. $LIM_A$ encompasses the temporal pole and adjacent regions of the ventral anterior temporal lobe (vATL)[151], which play a vital role in semantic cognition. Peelen and Caramazza[152] proposed that conceptual object attributes exhibit more abstract along the posterior–anterior axis of the ATL, reaching peak representation in the temporal pole, further supported by Devereux et al.[153]. The vATL is thought to act as a semantic hub that combines verbal and nonverbal conceptual knowledge to form a semantic concept[50–52,154,155]. For example, a study found that the vATL responds to auditory words, environmental sounds, and pictures, indicating that this region underpins multimodality semantic processing[155]. Furthermore, the vATL contributes to the representation of coherent concepts. An iEEG study observed that activity patterns of vATL were predicted by semantic similarity between objects[156]. An fMRI study revealed that similar objects elicited similar patterns of activation in the perirhinal cortex (the medial part of vATL)[157]. Similarly, the temporal pole is vital to semantic integration as proposed by a semantic hub model assuming that a semantic convergence zone in temporal pole for all kinds of concepts and semantics[158]. The temporal pole is related to the acquisition of new

conceptual knowledge[154], conceptual expansion[159] and semantic categories processing[158]. Studies found that the temporal pole contributes to creative thinking via semantic memory network[160]. An fMRI study on creative writing observed a right lateralized activation network including the temporal pole[88]. A study further found that experts in creative writing had reduced FC of the left caudate with left temporal pole, demonstrating that the disinhibition of the left temporal pole may contribute to excellent verbal creativity[161]. Taken together, although both conditions exhibited high uniformity of community assignments supporting multimodal semantic processing, the homogeneous activity of $LIM_A$ under semantic understanding condition may reflect limited multimodal engagement or a tendency to process semantically similar information.

$LIM_B$ had the lowest community overlap under both conditions, indicating that the activities in $LIM_B$ during the task were less uniform. The orbitofrontal cortex (OFC, corresponding to $LIM_B$) receives highly processed sensory and emotion information and projects to the medial striatum, mediodorsal thalamus, and prefrontal regions[151,162,163], which enables the OFC to guide the efficient matching of behavior to environmental contingencies through flexible encoding[163,164]. A clinical report had found the perfusion (hypoperfusion) in the OFC was significantly associated with lower creativity scores and the OFC may play a role in creativity via sensitivity to reinforcement and identification of emotions[165]. Structural neuroimaging studies suggest reduced OFC volume is associated with higher creative performance or divergent thinking abilities, indicating that thinner OFC is related to higher neural efficiency in supporting creative tasks[166,167]. Task-based fMRI studies on creative writing revealed that the brainstorming stage activated the bilateral OFC (BAs 10 and 11)[88], and the bilateral OFC (BA 10) was significantly activated when creative story generation was contrasted with uncreative story generation[168]. The primate OFC was found to respond to novel but not familiar visual stimuli[164]. These findings indicate that the OFC may be associated with unique blot generation by directing to new stimuli for further processing[169]. Time-window analyses further demonstrated that the OFC (BA 10) exhibited significant coupling with the DMN during later stages of alternate uses (vs. object characteristics), with no significant coupling in early stages[112]. Taken together, the OFC may contribute to creative cognition in a short temporal window by selectively responding to novel associations, which explains the less uniformly distributed communities of $LIM_B$ in the results.

Furthermore, TP was found to have the highest edge community similarity under semantic understanding condition. TP is located at the transition between the ventral and dorsal streams of visual processing while connects with the posterior dorsal lateral prefrontal cortex (DLPFC)[170]. The anatomical features of TP explain its functions of being the sites of semantic association and connected to the prefrontal cortex[171]. The superior temporal gyrus (STG) is heavily recruited during story/language comprehension[172,173], novel associations[170] and original AUT evaluations[174]. An fMRI study revealed that creative writing activated the posterior part of the left superior temporal region[88]. The result suggests that highly consistent semantic association processing occurred under semantic understanding condition.

Finally, we found that the $FPCN_A$, $FPCN_B$, $DMN_A$ and $DMN_C$ had the lowest edge community similarity under both conditions, while $DMN_B$ had the lowest edge community similarity under mental imagery condition and $FPCN_C$ under semantic understanding condition. These findings highlight a dynamic and flexible involvement of these networks during the demanding cognitive processes during the Thinking phase of creative writing.

The observation of low edge community similarity in these FPCN and DMN subnetworks suggests that their functional organization is more variable during the creative cognition, irrespective of the strategy. This may reflect the complex and diverse roles of FPCN and DMN in the creative cognition to support an interactive mechanism for broad and flexible access to specialized brain regions[175]. The DMN is composed of medial frontal, medial temporal, cingulate, precuneus, and inferior parietal cortical regions[109,176]. It enables complex forms of introspective thinking about stimuli not presented in the environment. The FPCN is composed of lateral

prefrontal and anterior inferior parietal regions[109]. It helps individuals achieve goals by dynamically adjusting and switching attention and responses[10,177,178]. Previous studies have shown that FPCN and DMN are central hubs in creative cognition by integrating complex information[9,98,112]. For instance, Kenett et al.[110] examined the community structures of resting-state functional brain networks, and found that in higher-creative individuals, DMN regions have increased participation coefficients, suggesting a key role in coordinating diverse information across communities. Finally, recent studies highlight the role of dynamic and complex coupling between the DMN and FPCN that realize creative thinking[11,102,104]. Taken together, the varied edge community distribution within FPCN and DMN suggests that they facilitate complex information processing and integration to support creative cognition.

$DMN_B$ consists of regions that overlap with the semantic control network, including the posterior middle temporal cortex (pMTG) and left ventral angular gyrus (AG). pMTG responds strongly to tool use and actions[179] and is widely implicated in tool-related semantics[180]. The left ventral AG showed stronger semantic responses when stimuli were pictures than written words, suggesting its role in conceptual identification of visual inputs[181]. A ventral AG site (bordering posterior temporal cortex) was found to show activity correlated with ATL and limbic areas[182]. Another major component of $DMN_B$ is the left superior frontal gyrus (SFG). A meta-analysis found visual mental imagery robustly engages SFG[183]. Mesial SFG is found to be importance for subsequent scene imagery task performance, suggesting its role in integrating and maintaining a scene representation[184]. Taken together, $DMN_B$ may play a special role in integrating semantic information related to objects and actions under mental imagery condition.

$FPCN_C$, referred to as the para-cingulate network in the work of Dadario and Sughrue[185], is composed of the posterior cingulate cortex (PCC) and precuneus and does not include a frontal component[186]. $FPCN_C$ is in a strategic position to facilitate the integration of internal and external stimuli and link it with previous knowledge to guide subsequent behavior. It plays a key role in working memory, especially updating it with new information[185]. $FPCN_C$ likely facilitates creative writing under semantic understanding condition by updating working memory through integration of external stimuli, internal states, and prior knowledge to generate contextually coherent output.

In summary, findings of Study 3 demonstrate shared and distinct patterns of large-scale network engagement during creative writing under two conditions. Across both conditions, several networks exhibited consistent coherence: $SMN_B$ plays a specialized role in auditory-related processing; DAN and SAL collaboratively support creative writing process by maintaining goal-directed attention and reorienting attention; $LIM_A$ supports multimodal semantic processing; $LIM_B$ may contribute to selectively respond to novel associations; FPCN and DMN support an interactive mechanism. Under mental imagery condition, $SMN_A$ facilitates sensorimotor simulations in creative cognition; $DMN_B$ may play a special role in integrating semantic information related to objects and actions. In contrast, under semantic understanding condition, $LIM_A$ may reflect limited multimodal engagement or a tendency to process semantically similar information; TP participates in highly consistent semantic association processing; $FPCN_C$ likely facilitates creative writing by updating working memory through integration of internal and external stimuli and prior knowledge to generate contextually coherent output.

## Limitations
Some limitations of this research should be noted. Our sample across all three studies primarily consisted of college participants. Although we implemented statistical controls for age and gender, future investigations should aim to replicate these results in more diverse populations with broader demographic characteristics and varying levels of writing expertise[161,187]. To ensure the ecological validity of the creative task, we used creative writing tasks for the measurement of creativity, thus it cannot purely detect the holistic relationship between mental imagery and creativity, as imagery engagement may vary across different domains of creativity[38]. Future research could benefit

from investigating the role of mental imagery in other forms of creative expression, such as visual arts or musical composition[35,37].

Secondly, semantic metrics provide a convenient, novel, objective, and ecologically valid way to reflect cognitive abilities, and previous studies have confirmed their capacity to reflect cognitive abilities[67,71,80,85,188]. However, to verify the reliability of the results, future studies should incorporate cognitive tests to examine individuals' semantic integration and reorganization abilities. Moreover, our semantic measure is based on a single semantic model and text corpus. Semantic integration relies on the Chinese-BERT-wwm-ext model, and semantic network robustness and creative writing performance ratings depend on the Word2Vec model. The choice and scope of text corpora can introduce biases, as different corpora vary in the ability to capture human performance[189,190]. A more robust approach would be to integrate multiple semantic models and corpora.

Additionally, we constructed edge communities of the brain, which facilitates the investigation of the overlapping brain functional organization of creative cognition with the involvement of imagery. However, there are certain limitations in revealing mechanisms. Firstly, edge communities are the result of clustering based on the fluctuation patterns of edges, thus they do not construct how information flows on edges. Moreover, since mental imagery and creative cognition are both high-order cognition[97,191], it is not possible to categorically attribute a binary causal classification to the brain regions that played a key role during the task. For example, it cannot be entirely assumed that the lowest similarity of DMN's edge communities is simply because it plays a role in information integration during creative activities, while ignoring the possibility that it may also be involved in the top-down construction of imagery[192]. Future research should, on one hand, expand the analysis based on the edge-centric functional networks approach, analyzing edge networks and edge pathways; on the other hand, it should correlate brain network indicators with behavioral and other modal indicators to further reveal the cognitive processes represented by edges[115].

## Conclusion
In conclusion, our findings offer key insights into how mental imagery supports creative cognition, particularly through semantic memory and the distributed architecture of brain network edge communities. By applying advanced semantic analysis and edge-centric brain functional network approaches, we provide a fresh perspective on the intricate connections between mental imagery and creativity, paving the way for future research on the broader cognitive and neural implications of mental imagery in creative cognition.

## Methods
### Participants
In Study 1109 college students were recruited for completing the Psi-Q scale and a creative writing task. Nine participants were excluded for errors on the screening questions, one for an excessively brief response time (576 s), and one for an outlier Psi-Q touch score, which fell outside the 25th–75th percentile distribution, with the score being 2, resulting in a final sample of 98 (78 females; mean age 21.63 years ± 2.16). Study 2 involved 68 participants (52 females; mean age = 21.12 years ± 1.66) in a within-subjects design. Study 3 (fMRI) recruited 30 students, with one excluded due to scanning issues, leaving 29 for analysis (21 females; mean age 21.48 years ± 2.33). All procedures were approved by the institutional review board of Southwest University, and participants were healthy with no history of neurological or psychiatric conditions, no metal implants, and no claustrophobia. Informed consent was obtained, and participants were compensated upon completion. All ethical regulations relevant to human research participants were followed.

### Procedure
**Study 1**. Mental imagery vividness was assessed using the Plymouth Sensory Imagery Questionnaire (Psi-Q)[86], which required participants to form mental imagery based on given items and rate each image on a scale from 1 (no imagery at all) to 5 (imagery as vivid as reality). The

questionnaire encompasses seven sensory dimensions: vision, sound, smell, taste, touch, bodily sensation, and emotional feeling, with five items per dimension. Creativity was measured using a short story generation task. Participants were given a three-word prompt, stamp-letter-send[80], and asked to include all three words when typing a short creative story about 4–6 sentence, 50–100 words in length. No time limit was imposed (Fig. 5A).

**Study 2.** Participants were tasked with creative writing under both mental imagery and semantic understanding conditions. Study 2 was structured to compare the impact of these two cognitive strategies on the creative writing process (Fig. 5B).

Study 2 commenced with an instructional phase. Participants were introduced to the concepts of mental imagery or semantic understanding, with guidance on simulating sensory experiences for mental imagery condition and analyzing concepts for the semantic understanding condition. Examples were provided to clarify these strategies, such as evoking the sensory experience of "cutting a lemon" or focusing on the functional aspects of "cutting a lemon". Participants were also informed of the creative writing requirements: to create a 3–5 sentence story based on 2–3 cue words, which could include characters, time, place, events, and plot. After the introduction, they rated, on a scale from 1 to 11, the vividness of their mental imagery and the clarity of their semantic understanding regarding lemon-cutting, as well as their comprehension of the concepts of mental imagery and semantic understanding.

The practice phase followed, where participants applied the corresponding strategy to a creative writing exercise. They were given a text with

two sections: an initial event statement followed by a description that either detailed the sensory experience of the event or provided a semantic explanation of the event. Participants were asked to craft a 3–5 sentence story based on 2–3 cue words from the event, utilizing the strategies they had just practiced. For mental imagery condition, participants were directed to generate the sensory experiences (aligning with the seven Psi-Q dimensions) related to the 2–3 cue words and to craft stories from these images. For semantic understanding condition, participants were prompted to reflect on the presented words' meanings, principles, functions, and values, and to write based on the comprehension. After writing, they rated the vividness of their mental imagery or the clarity of their semantic understanding for the cue words, as well as their application of the strategies in their writing, using a 1 to 11 scale.

In the formal writing phase, participants completed 10 creative writing tasks for each of the conditions, totaling 20 tasks. Each condition featured a predetermined sequence of 10 unique events, with participants finishing all tasks in one condition before proceeding to the other. The sequence of conditions and event materials is balanced across participants.

**Study 3.** Participants completed 10 creative writing tasks for mental imagery and semantic understanding conditions, totaling 20 tasks (Fig. 6A). The order of the two conditions was randomized but fixed across participants. Event materials were divided into two sets: Set A consisted of the mental imagery version of 10 events and the semantic understanding version of another 10 events; Set B was the reverse. The events mirrored those from Study 2, but text length was limited to 300 characters to minimize participant fatigue and distraction. Before MRI

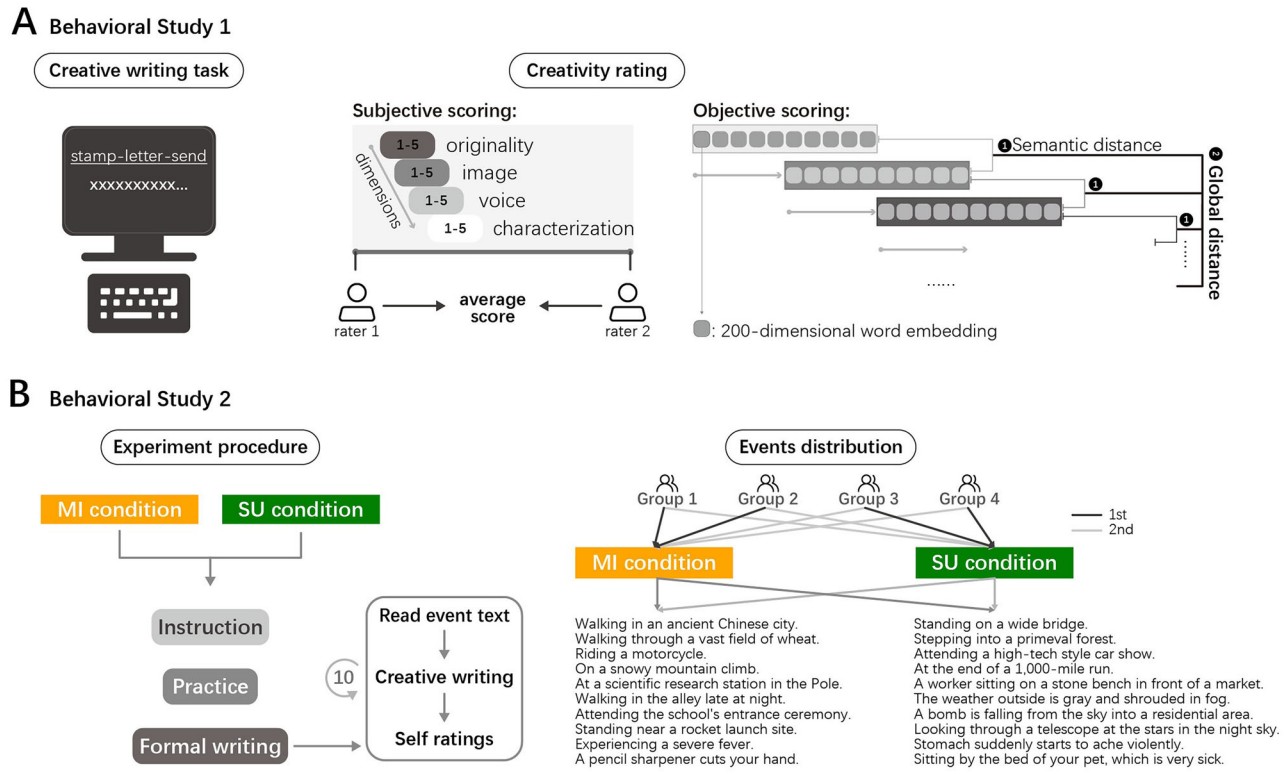

**Fig. 5 | Procedures of creative writing task. A** Creative writing task of Study 1. Participants engaged in creative writing based on three given words and input the text into the computer. The creativity scores for creative writing here were assessed in two ways: subjective and objective scoring. Subjective scoring required two raters to score on a 1–5 Likert scale across four dimensions and sum the scores of dimensions, followed by averaging the scores of the raters. Objective scoring employed global distance quantification. The text was cleaned and organized into a high-dimensional matrix of 200-dimensional word embeddings. A sliding window size of 10 with a sliding step size of 5 was set to calculate the semantic distance between adjacent windows. Finally, the average of all semantic distances yielded the global distance. **B** Creative writing task of Study 2. The left panel illustrates the experimental procedure for each condition for a participant. The right panel illustrates how the presentation sequence of conditions and event materials is balanced across participants. Note that participants were randomly assigned to four groups, with each participant completing 10 events of mental imagery condition's creative writing tasks and another 10 events of semantic understanding condition's creative writing tasks. MI mental imagery, SU semantic understanding. Icons were obtained from the open-access web-based software (ICONFONT, https://www.iconfont.cn).

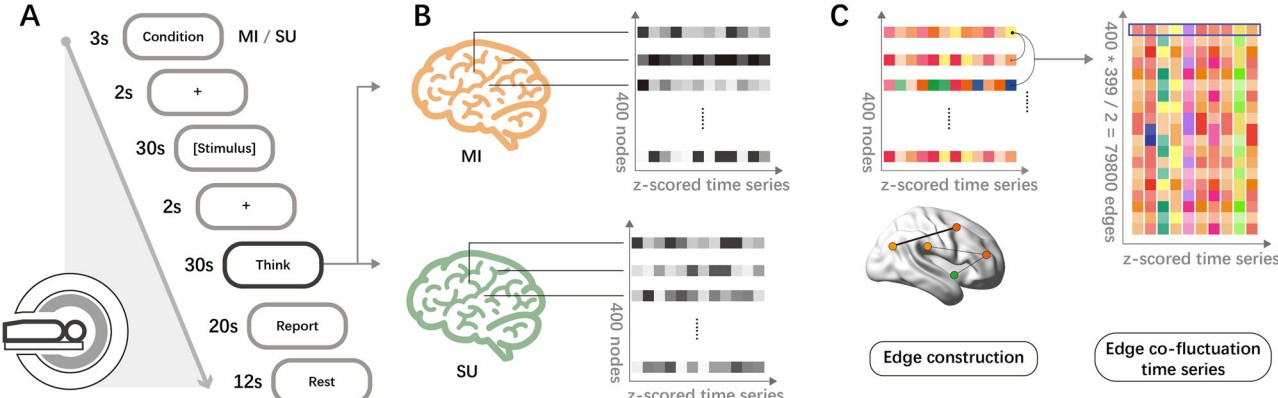

**Fig. 6 | Study 3 procedure and edge construction. A** The flowchart of a single trial. The creative writing condition is presented for 3 s. After the 30-s presentation of the event description corresponding to the condition, participants need to think and verbally report a creative short story based on 2–3 cue words presented on the screen (words from the described event) and strategies of corresponding condition. Finally, a 12-s rest period was followed to allow the participants' brain activities to return to baseline levels. **B** Extracting the blood oxygen level-dependent (BOLD) signals during the Think phase under two creative writing conditions and connecting them separately. **C** Subtracting the z-scored BOLD time series from the two conditions to

represent brain activity specific to the creative cognitive process involving imagery encoding. Edges were constructed by multiplying the z-scored BOLD signals of each brain region pair at a time point. To create edge co-fluctuation time series, the BOLD signals of all brain regions were multiplied at each time point. The fMRI scanning icon and brain atlas icons in (**B**) were sourced from ICONFONT. The brain template in (**C**) was obtained from the BrainNet Viewer toolbox (www.nitrc.org/projects/bnv/)[203]. All graphical elements were post-processed for color adjustment and optimization. Time series illustrations were manually drawn.

scanning, the experimenter detailed the procedure, concept definitions, and requirements, highlighting the importance of focused attention and strategic writing over text summarization.

### Creativity assessment of creative writing
**Subjective scoring.** In Study 1, two creativity experts rated the originality, image, voice, and characterization of the participants' stories on a 1–5 scale[193] (Fig. 5A). The method has been validated in prior research[87]. Study 1 utilized the average of the sum scores across these four dimensions as the assessment for creative writing performance. Inter-rater reliability was calculated using Intraclass Correlation Coefficient (ICC), with an ICC(C,K) = 0.730, indicating good agreement between raters. The average of the sum of ratings from two raters was taken as the subjective score for creative writing performance.

**Objective scoring.** We applied global semantic distance to assess the objective score of creative writing performance in Studies 1 and 2. Semantic distance is typically calculated by converting words into high-dimensional vectors or embeddings and measuring the cosine distance between them. Previous studies have shown that greater semantic distance between responses and cue words in divergent thinking tasks is associated with higher associative abilities[67,78,114]. Global semantic distance is a reliable and valid measure of creativity in creative writing[87]. It has been validated against individual differences in creative behavior, including creative achievements and personality, and has been shown to predict individual differences in creativity and behavioral performance. Additionally, its reliability was tested against subjective creative writing scores, revealing a significant correlation.

The calculation of global semantic distance involved several steps. First, meaningless punctuations were removed, and the text was tokenized using the jieba Chinese tokenizer (https://github.com/fxsjy/jieba). Next, 200-dimensional word embeddings were obtained from the Tencent AI Lab Embedding Corpus for Chinese Words and Phrases (https://ai.tencent.com/ailab/nlp/en/download.html). This corpus includes 8.82 million words and phrases and was pre-trained using the Directional Skip-Gram (DSG) model[194]. The text was then divided into multiple windows of 10 words each, with word vectors averaged within each window to produce a single averaged vector. The semantic distance between adjacent windows was calculated using a sliding window of 5 words, defined as 1 minus the cosine similarity between adjacent vectors. Finally, the semantic distances across all

windows were averaged to determine the global semantic distance[87] (Fig. 5A).

### Semantic features
**Semantic integration.** Semantic integration was assessed using the Divergent Semantic Integration (DSI) indicator developed by Johnson et al.[80], which is based on semantic distance to capture the representation of distantly associated concepts and their integration into a story, reflecting overall coherence. We employed the Chinese BERT pre-trained model with Whole Word Masking (Chinese-BERT-wwm-ext) for generating word embeddings and calculating semantic distance. This model was released by the Joint Laboratory of HIT and iFLYTEK Research (HFL)[195] (https://gitcode.com/ymcui/chinese-bert-wwm/overview). Bidirectional Encoder Representations from Transformers (BERT) has become a widely used language model in natural language processing, utilizing large-scale unannotated training data to produce contextually relevant word embeddings[195].

The written text was first segmented by sentences, and then Chinese-BERT-wwm-ext was used to extract features for each sentence. Specifically, the representations from the sixth and seventh hidden layers were extracted as feature vectors for each word in the sentences, as using all layers would have been computationally expensive, and these two hidden layers were considered sensitive to syntactic and semantic information[196]. The average cosine distance between each pair of words across the two hidden layer representations served as the semantic integration indicator for the entire story (https://osf.io/ath2s/).

**Semantic network robustness.** Cognitive flexibility, a core component of cognitive control, refers to the ability to adaptively shift goals and task sets in response to changing environmental demands[197], which is essential to creativity[198]. Recent advances in semantic network analysis, pioneered by Kenett and colleagues, have introduced novel computational techniques to quantify cognitive flexibility through robustness analysis of semantic networks[71,81,82], with networks comprising nodes (semantic memory or concepts) and edges (semantic similarity).

Semantic network robustness examines the flexibility of semantic memory networks by progressively removing edges with strengths below a certain threshold[199]. Highly creative individuals have more flexible networks that are more robust to attack[71]. Greater robustness indicates enhanced flexibility and more effective reorganization of information within the

network. Previous studies on cognitive flexibility in creativity often relied on indirect measures, such as task-switching[67,79]. In contrast, semantic network robustness provides a direct measure of semantic memory and creative flexibility. The analysis involved the following steps:

*Network construction.* The jieba Chinese tokenizer was used to segment the text produced by the participants. A 200-dimensional word vector was obtained from the corpus provided by Tencent AI Lab. A fully connected semantic network was constructed based on words and word vectors, where nodes represented words and edges represented the Pearson correlation between word vectors (Fig. 2C, D). In Study 2, we controlled for the influence of word count under two conditions (See Supplementary Method).

*Network percolation analysis.* A set of thresholds was defined for percolation analysis of the semantic network. The threshold range was from 0 to 1 with a step size of 0.0125 (i.e., threshold resolution). Edges with weights less than the threshold were iteratively removed from the semantic network at thresholds with a step size of 0.0125, and the size of the giant component (the largest connected group of nodes) was calculated. If the size of the giant component was less than 3 at a certain threshold, the iteration stopped. Finally, the percolation integral was calculated for each participant.

$$PI = \int_{start\_\_TH}^{end\_TH} GC(x)dx = \sum_{TH-start\_\_TH}^{end\_TH} GC(TH) * TH\_res$$

PI is the percolation integral, GC is the giant component, TH is the threshold value, start/end_TH are the initial and final threshold values, and TH_res is the threshold resolution.

*Effect of noise analysis.* To evaluate the statistical significance of semantic network robustness, a noise analysis was performed to assess the impact of introducing noise to each of the network's edge weights, aligning with the work of Kenett, Levy, et al.[71]. If the introduction of noise significantly alters the relationships between PI and other variables (e.g., creative writing ratings), it would indicate that the percolation process is not statistically robust. The PI was calculated over 500 realizations of the percolation analysis. In each realization, a Gaussian noise was added to the network with a mean value of zero and a variable SD ranging randomly between the minimum and maximum SD. The minimum SD was computed as the reciprocal of the total number of edges[71], while the maximum SD was set to ten times the minimum SD, providing a range of noise levels.

*Link shuffling analysis.* Link shuffling analysis was done to examine the structure of the network and its effect on the percolation process. the analysis was to examine whether the semantic network robustness under two conditions were caused by the structure of the network and not the weights of the edges in the network. In the shuffling process, two edges in the network were randomly chosen and exchanged, and the PI was calculated. In each iteration, 80% of the total number of edges were chosen to be shuffled. We iterated the shuffling process for 20 times, which was determined to facilitate sufficient shuffling to achieve a probability of ≥0.95 for each edge to be shuffled at least once. The average PI across all iteration was calculated to obtain the final PI for each network.

## Mediation analysis

After an initial exploration of the relationship between mental imagery and creative writing performance, further investigation examined the potential mediating role of semantic features, i.e., semantic integration and semantic network robustness. Indirect effects were assessed using the bootstrapping method, with 1000 bootstrapped samples computed for each analysis. The 95% confidence interval (CI) was computed by determining indirect effects at the 2.5th and 97.5th percentiles. Mediation analyses were performed in jamovi (https://www.jamovi.org/).

## MRI data acquisition and preprocessing

The MRI scanning imaging was conducted with a 3-T Siemens Prisma scanner (Erlangen, Germany) at Southwest University. Study 3 utilized magnetization prepared rapid acquisition gradient echo (MPRAGE) to acquire structural images, with scan parameters including: repetition time (TR) = 2530 ms, echo time (TE) = 2.98 ms, flip angle (FA) = 7°, field of view (FOV) = $256 \times 224$ mm$^2$, voxel size = $0.5 \times 0.5 \times 1.0$ mm$^3$, slice thickness = 1.0 mm. Task-based fMRI data were collected using a gradient-echo planar imaging (EPI) sequence, with scan parameters of TR = 1000 ms, TE = 30 ms, FA = 73°, FOV = $195 \times 195$ mm$^2$, voxel size = $2.5 \times 2.5 \times 2.5$ mm$^3$, slice thickness = 2.5 mm.

Preprocessing and denoising of the task-based fMRI data were done using dpabi (version 8.1)[200] toolbox in MATLAB 2021a (https://matlab.mathworks.com). The steps included: (1) head motion realignment; (2) co-registration: registering the functional images to the T1-weighted images and standardizing to the Montreal Neurological Institute's 152 brain template (MNI152), and resampling the voxel size to $2.5 \times 2.5 \times 2.5$ mm$^3$; (3) denoising steps on the functional imaging to reduce the impact of non-neuronal fluctuations. Regression covariates included five noise principal components from white matter (WM) and cerebrospinal fluid (CSF), Friston's 24 head motion parameters, and the first-order linear effect. Then, bandpass filtering (0.01–0.1 Hz) was applied to reduce the effects of low-frequency drift and high-frequency physiological noise; (4) smoothing the functional images with a 5-mm full-width half-maximum Gaussian kernel to improve the blood oxygen level-dependent (BOLD) signal-to-noise ratio and to reduce anatomical differences caused by inaccuracies in inter-subject registration.

## Edge-centric functional network Analysis

**Edge graph construction.** Initially, the BOLD signals during the Think phase of the mental imagery and semantic understanding writing conditions were extracted. The time series of BOLD signals representing the creative writing process under mental imagery and semantic understanding conditions were generated by concatenating the z-scored BOLD signals from 10 trials for 400 brain regions[201] (Fig. 6B). For parcellation preprocessing, please view Supplementary Method.

Subsequently, we constructed the regional co-fluctuation patterns by multiplying the BOLD signals of all brain regions at each time point, i.e., edges (Fig. 6C). For each pair of regions, for example, $i$ and $j$, their standardized time series $z_i$ and $z_j$ were multiplied point by point to obtain a new vector of length T (number of time points), where each number represents the magnitude of the instantaneous co-fluctuation between brain regions $i$ and $j$ at that time point. If both regions $i$ and $j$ increase in activity at a certain time, the corresponding value in the vector indicates a positive co-fluctuation. If one region increases and the other decreases, the corresponding value indicates a negative co-fluctuation. If one region increases or decreases in activity while the other remains near baseline levels, the corresponding value indicates a co-fluctuation close to zero (Fig. 6A). This operation was to obtain a set of co-fluctuation (edge) time series for brain regions. Then we averaged edge time series across subjects to obtain a representative matrix for further analyses.

**Edge community analysis.** Edge community refers to edges that have similar fluctuation patterns over time and are marked as belonging to a community, potentially reflecting communication patterns between different brain regions. To detect the communication patterns of brain regions during creative writing under both conditions, brain edge communities were generated through clustering, and their distribution was analyzed across Yeo-Krienen 17 networks[202]. Mapping these edge communities onto the 17-networks atlas means that each edge belongs to a single community, while the nodes connected by the edge may belong to multiple communities. The process aided in understanding the association between the high-dimensional edge communities and traditional brain networks (Fig. 7C, Supplementary Figs. 7 and 8). The following steps were taken for the analysis:

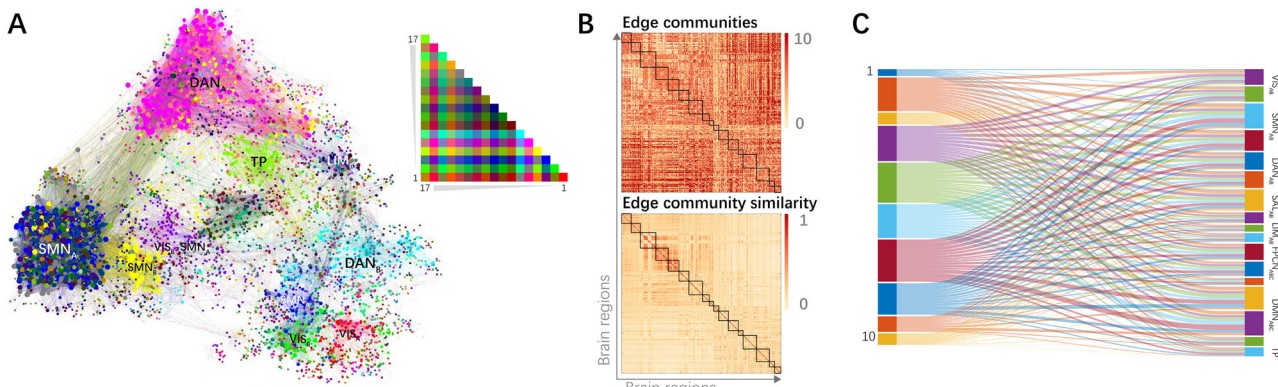

**Fig. 7 | Edge functional connectivity and edge community construction.**
**A** Visualization of the edge functional connectivity (eFC) under mental imagery condition using a force-directed layout. The correlation matrices of all participants' time series are averaged to form a single matrix representing eFC, which is visualized with a threshold of 0.95. Each point in the graph represents an individual edge, and the color of the point is determined by the brain network to which the edge's stub nodes belong. **B** Matrices corresponding to the edge community and edge community similarity for the 400 brain regions of the Yeo-Krienen 17-networks atlas ($k = 10$) under mental imagery condition. The x-axis and y-axis represent the 400 brain nodes. In the Edge communities matrix, the position $i$-$j$ indicates the

community number (ranging from 1 to 10) to which the edge formed by brain nodes $i$ and $j$ belongs. In the Edge community similarity matrix, the position $i$-$j$ represents the degree of similarity between the edge communities to which brain nodes $i$ and $j$ belong. **C** Sankey plot illustrating the correspondence between the communities to which edges belong and the brain networks where the connected nodes reside ($k = 10$) under mental imagery condition. The plot demonstrates that non-overlapping edge partitions result in overlapping nodal partitions. VIS visual network, SMN sensorimotor network, DAN dorsal attention network, SAL salience network, LIM limbic network, FPCN frontoparietal control network, DMN default mode network, TP temporal-parietal network.

*Detection of edge communities in brain networks.* Clustering was directly applied to the time series of edges rather than first generating a functional connection matrix for the edges and then reducing its dimensions and clustering. Following established methodological standards[115,120], we applied k-means clustering with squared Euclidean distance (5000 iterations) to identify edge communities, varying the number of communities, $k$, from $k = 2$ to $k = 20$. Our main analyses report the average patterns across all $k$ to demonstrate the robust spatial distribution of edge communities.

*Edge community overlap within network.* Normalized entropy was used to measure the level of overlap of edge communities for a node in a given brain network. It measures how a node in a brain network is distributed across different edge communities. Entropy is typically used to describe the uncertainty or disorder of a system. In this context, normalized entropy indicates the diversity of the distribution of edges in communities within a brain region. The normalized entropy ranges between 0 and 1, with values close to 0 indicating that most nodes in a given network are in a few edge communities, while values close to 1 indicate that nodes are distributed across many edge communities. Thus, the closer the normalized entropy is to 0, the less overlap between communities, and the closer it is to 1, the more overlap.

*Edge community similarity within network.* The similarity between edge communities in which nodes i and j of a specific brain network are located was compared. By assigning nodes to individual communities, the communities to which nodes belong can be rearranged into an N × N (N being the number of brain regions) matrix X, where element $x_{ij}$ indicates the edge community allocation between nodes $i$ and $j$. Column $i$ of matrix X, denoted as $x_i = [x_{1i}, \ldots, x_{Ni}]$, represents all the edge community labels that node $i$ participates in. Note that element $x_{ii}$ is left empty since self-connections are not considered. The similarity between vectors $x_i$ and $x_j$ was then calculated, defined as the proportion of elements with the same community label in the two vectors. The average similarity of edge communities within each brain network was calculated, resulting in a N×1 matrix representing the average similarity of each node with all nodes within the same brain network (Fig. 7B).

MATLAB code for edge graph construction and community analysis is publicly available[120] (https://github.com/brain-networks/edge-centric_demo).

## Statistics and reproducibility
All statistical tests used and sample sizes details for both behavioral data and fMRI data are described in the sections of the "Results" and "Methods."

## Reporting summary
Further information on research design is available in the Nature Portfolio Reporting Summary linked to this article.

## Data availability
The data and material used in this study are available from the corresponding author (J.Q.) upon reasonable request. The source data behind the figures in the paper can be found in Supplementary Data 2.

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

## Acknowledgements
This research was supported by the Chongqing Social Science Planning Project (2022PY14), the West China Institute of Children's Brain and Cognition of Chongqing University of Education (XW2022021) and the National Natural Science Foundation of China (32471096). We would like to thank all the participants and members in semantic group in Prof. Qiu's Lab.

## Author contributions
Jing Gu: Investigation; conceptualization; data curation; methodology; formal analysis; visualization; writing - original draft; writing - review & editing. Xueyang Wang: Investigation; conceptualization; supervision; writing - review & editing. Cheng Liu: Conceptualization; methodology. Lin Yang: Investigation; data curation. Jiaxin Fan: Investigation; data curation. Jiangzhou Sun: Supervision; writing-review & editing. Yoed Nissan Kenett: Writing - review & editing; supervision. Jiang Qiu: Conceptualization; funding acquisition; project administration; resources.

## Competing interests
The authors declare no competing interests.
