## [Transparent Peer Review file · Communications Biology]

Cognitive and neural mechanisms of mental imagery supporting creative cognition

Corresponding Author: Professor Jiang Qiu

Version 0:

Reviewer comments:

Reviewer #1

(Remarks to the Author)

What are the major claims of the paper?

This paper claims that mental imagery (mainly of the tactile type) enhances creative writing performance, due to the fact that it supports semantic memory reorganization and representation. Moreover, mental imagery's impact on creative writing performance is promoted by semantic integration (representational capability) and semantic network robustness (reorganizational flexibility), suggesting that mental imagery facilitates access to diverse and distant concepts and their flexible manipulation. Finally, in order to analyze the dynamic processes of creative cognition through fMRI, the authors adopted an edge-centric functional networks approach, which resulted in the following being claimed as neural correlates of creative cognition with imagery encoding: the sensorimotor network (SMN, acting as a processing hub), the dorsal attention (DAN) and salience (SAL) networks coordinating resource allocation, the frontoparietal control network (FPCN) and default mode network (DMN), which dynamically integrate complex information.

Are they novel and will they be of interest to others in the community and the wider field?

The part of this work that presents a more innovative approach is the fMRI study, in which edge-centric functional connectivity is deployed as a tool to understand neural correlates of mental imagery during creative writing tasks. The emphasis of the tactile domain is also interesting. Finally, employing semantic integration and percolation analyses of semantic memory networks is an elegant methodological approach that can be appreciated by the scholarly community (although used before in the field by Kenett – amply cited by the authors)

If the conclusions are not original, it would be helpful if you could provide relevant references. Is the work convincing, and if not, what further evidence would be required to strengthen the conclusions?

The authors declare, as a limitation of the findings: "Future research could benefit from investigating the role of mental imagery in other forms of creative expression, such as visual arts or musical composition" (line 883). For this reason, I would like to suggest that the previous literature review includes some works that partially fill that gap, such as Wong SSH, Lim SWH (2017) Mental imagery boosts music compositional creativity. PLoS ONE 12(3): e0174009. <https://doi.org/10.1371/journal.pone.0174009>.

The inclusion of De Pisapia, N., Bacci, F., Parrott, D. et al. Brain networks for visual creativity: a functional connectivity study of planning a visual artwork. Sci Rep 6, 39185 (2016). <https://doi.org/10.1038/srep39185> could also provide further context for networks involved in creativity in visual arts, in the literature section in lines 127 and following (De Pisapia published several other contributions on this topic).

On a more subjective note, do you feel that the paper will influence thinking in the field? Please feel free to raise any further questions and concerns about the paper.

The paper has the potential to influence further research. I don't have major concerns in that regard, but rather a suggestion so that the authors don't miss the opportunity for widening the audience. In the paper's bibliography there is evidence that the authors considered a wider interdisciplinary angle. For example, the inclusion of more considerations from the point of view of neurobiology could open up the discussion in fruitful directions. As an acknowledgment of Nature's commitment to

welcome interdisciplinary voices, also briefly crediting phenomenology and previous philosophy as seminal in sensory studies would have been fruitful (I have appreciated the definition of touch through poetic means – lines 690-693).

We would also be grateful if you could comment on the appropriateness and validity of any statistical analysis, as well the ability of a researcher to reproduce the work, given the level of detail provided.

The level of detail in describing the methodology utilized seems quite comprehensive (which would also help with the replication of the experiments) the following observations (which relate more to the methods than the statistical analysis) need addressing:

- I would like to begin by commending the author for gathering such an extensive number of participants. However, the pool of subjects is comprised of majority female students in their 20s. A more varied and balanced population could have yielded different results, probably also based on variations in expertise, degree of studies and age.

- We don't know enough about the expertise of the subjects. Some may be training to become writers, or some may be using creative writing in the context of another creative profession, such as performing arts, visual arts or architecture (imagining and writing about the specificities of a movie set design, that gets built from the description, as an example). Such individuals may be using mental manipulation of semantic descriptions or images frequently in their practice. The role of expertise in creative tasks has been established, since expertise changes brain structures and functionalities in the specific brain areas that are used in a specific task [for examples, see Pascual-Leone, A. The brain that plays music and is changed by it. *Annals of the New York Academy of Sciences* 930, 315–329 (2001); Maguire, E. A., Woollett, K. & Spiers, H. J. London taxi drivers and bus drivers: a structural MRI and neuropsychological analysis. *Hippocampus* 16, 1091–1101 (2006)].

- Lines 276-286 Task instruction seem very complex, especially for undergraduate students, with a high chance of non-compliance of execution due to the complexity of task. I suggest providing more details regarding the training phase to clarify how ambiguity due to poor task comprehension was avoided.

- Line 598 please correct the grammar – verb missing?

- What is the role of attention during task execution? Have you recorded any fluctuations and moments of loss of focus? It seems that 20 repetition of such a demanding task may generate decrease in focus and “creative fatigue”. Did you notice a difference in the first half of the set of tasks and the second half, in term of performance by the each individual subject?

Reviewer #2

(Remarks to the Author)

The authors present a series of three studies exploring the relationship between mental imagery, semantic memory, and creative writing. The authors provide behavioral results that demonstrate that the influence of mental imagery on creativity is mediated by semantic features of the creative outputs, and that mental imagery is as a more effective strategy for facilitating creativity compared to semantic elaboration. Using task-based functional connectivity and network modeling, the authors identify properties of whole-brain functional connectivity that characterize mental imagery vs semantic creative strategy.

The focus on the relationship between mental imagery, semantic processing, and creativity is well-motivated and offers potential for meaningful insights into the neurocognitive underpinnings of creativity. Methodological strengths include the unique datasets, the use of edge functional connectivity modeling and the comparison of the results with a well-known reference brain atlas. I acknowledge that I am not deeply familiar with the edge functional connectivity modeling (eFC) approach adopted here, so I am unable to evaluate that aspect of the analytic pipeline. However, I have concerns regarding the clarity of the background & rationale, certain methodological choices, and the overall clarity and organization of the writing. I elaborate on these concerns in the detailed comments below.

Major Comments

Introduction

1. The introduction includes key relevant content but is challenging to follow in places. To improve its structure and clarity, I recommend reorganizing the content to prioritize the hypotheses and their rationale before describing and motivating specific methods (e.g., eFC) or construct operationalizations (e.g., semantic integration vs reorganisation).
2. To improve the clarity of the theoretical grounding of the study. I recommend addressing the following:
 - Define creative cognition in terms of its behavioral manifestations (e.g., painting, poetry, dance choreography) and its relevance to the hypotheses.
 - Clarify the distinction between modality-specific and supra-modal semantic representations and how they relate to creativity and mental imagery, as well as the brain networks referenced.
 - Elaborate on what is meant by “creative cognition being embodied.” A deeper dive into the embodied cognition framework is needed to clarify the link between creativity, mental imagery, and semantic processing.
 - Provide definitions of the executive control network and the default mode network, including their subcomponents (e.g., FPCNa and FPCNb), and explicitly link them to the cognitive constructs of interest (creativity, mental imagery, and semantic memory).
3. The rationale for the study's hypotheses and methodological choices would benefit from more clarity or expansion:

- The hypothesis that creative cognition supports “overlapping brain functional organization for information processing” lacks a clear rationale. Additionally, the importance of testing overlapping vs. non-overlapping functional architectures is not clear. Why is this question critical for the study, and for the field?
 - The choice of edge-centric functional connectivity (eFC) as the primary analytical approach is unclear. Why is this method particularly suited to the study’s research questions? Moreover, throughout the introduction, it was unclear to me whether the eFC would be derived from task-based fMRI or whether eFC metrics would be related to behavioral measures obtained outside the scanner.
 - The motivation for the operationalizations of “semantic integration” as semantic representation and “cognitive flexibility as network robustness” is not clear. Moreover, integration and representation are slightly different components of semantic processing and should not be equated.
4. Some statements are vague, making it difficult to grasp the intended meaning:
- “Body movement and posture play a direct role in creative cognition.” Do you mean that many creative tasks involve bodily movement? It is unclear how whole-body movement relates to tasks like generating creative jokes. Providing specific examples could help clarify your argument.
 - “We utilized the edge-centric functional networks approach to analyze blood oxygen level-dependent (BOLD) signal fluctuations of creative cognition with imagery encoding involvement to capture brain organization distribution patterns at a single-frame resolution.” What is meant by “BOLD of creative cognition”? Are you referring to functional connectivity during task-based fMRI? What is “imagery encoding involvement”? What do you mean by single frame resolution?

Methods

5. The use of a subtraction approach in this study does not seem well-justified. The task conditions (semantic strategy and mental imagery strategy) appear to engage distinct cognitive processes rather than represent additive components of the same cognitive process. A more informative approach would have been to compare the characteristics of the brain networks estimated under each condition separately, highlighting the unique patterns associated with each strategy rather than relying on a subtraction-based comparison.
6. The decision to set the number of communities a priori to 10 is not adequately justified. Why was this specific number chosen, and how does it align with the data or any theoretical considerations? To strengthen the analysis, the authors should provide evidence that $N=10$ is the optimal number of communities—such as through community quality metrics (e.g., modularity)—or demonstrate that their results are robust across a range of community numbers.

Results

7. The Results section includes unnecessary repetition of methodological details and would benefit from more concise descriptions.
8. The methods provided in the subsection titled “Imagery strategy enhances creative writing performance.” would be better suited in the Methods section.
9. The description of self-reported measures is unclear. For example, the Likert scale extremes are not defined, making it difficult to interpret statements like: “One-sample t-test analysis revealed that mental imagery vividness and usage, as well as semantic understanding clarity and usage, were significantly above the average score of 6 (all p 's < .001).”

Discussion

10. The results should be discussed with reference to the hypotheses stated in the introduction. As currently written, there is no reference to the hypotheses in the discussion.
11. Several sentences in the discussion merely restate results without interpreting them and would benefit from conciseness. For example: “Furthermore, DANA showed both highest edge community overlap and similarity, indicating extensive similar information processing in DANA during creative writing.”
12. Some interpretations are not fully supported by the data. For instance:
- “We further found the vividness of touch imagery influenced creative writing performance through two semantic features, i.e., semantic integration and semantic network robustness, which could be interpreted as vivid touch imagery promoting creative writing performance by representing novel concepts and flexibly manipulating these concepts.” No measure of haptic features of semantic representations or the novelty of concepts was explicitly assessed.
 - “While both strategies lead to rich semantic representations, flexible manipulation of these concepts occurs only under the mental imagery condition” The authors did not provide direct evidence demonstrating that semantic reorganization is unique to the mental imagery condition, nor did they explicitly measure the richness of semantic representations.
13. The interpretations that rely on speculative reasoning and reverse inference would benefit from the use of cautious, evidence-aligned language, such as “the results are consistent with the possibility that.” to more clearly differentiate between speculation and what the results demonstrate. For example, “The cooperation of both networks indicates that participants were able to flexibly switch their attention between internal and external resources to search for results, which support the dual-process model of creativity, positing an interaction (or coupling) between two latent cognitive modes (i.e., spontaneous/implicit thinking and deliberate/explicit thinking) during the creative cognition”). This interpretation is purely based on previous associations between the two networks and internal vs external attention. It is possible that those regions performed different functions during the tasks used in this study, and, therefore, the current results do not provide direct support for the dual process model of creativity.
14. Some statements are vague, making interpretation difficult. For example:
- “DAN potentially executed creative writing tasks through internal attention processes involving substantial single information processing.” What does “substantial single information processing” mean?
 - “Mental imagery engagement may function as an information processing mechanism, with working memory encoding

sensory information to facilitate flexible information processing.” This statement is too broad, as all cognitive functions could be said to involve “information processing mechanisms.”

• “The highest edge community similarity indicates that the emotion processing within LIMA was relatively singular.” What does “relatively singular” mean?

Minor Comments

15. “Which required participants to form sensory images based on given items and rate each image.” Specify that you mean mental images or images in the mind’s eye, rather than physical images, for precision.

16. Clarify that the participants completed the 20 creative writing tasks during fMRI scanning.

17. Expand on how the “200-dimensional word embeddings from Tencent AI Lab’s Chinese term embedding corpus” were derived.

18. “The cosine distance between each word in these two hidden layer representations was calculated.” Do you mean the cosine distance between each pair of words? Please clarify.

19. Define the term “giant component” and explain what the “percolation integral (PI)” quantifies. Additionally, clarify how the existence or lack of a “dramatic change” in PI was determined or quantified.

20. Avoid the use of bar graphs for presenting data, as they can obscure important details about distributions (see Newman & Scholl, 2012).

21. “Community detection approaches applied to brain networks aim to find densely interconnected sets of nodes, leading to the notion that the brain is organized in a modular manner.” It is important to note that the ability to apply these models to brain data does not necessarily imply that the brain functions modularly. Additional empirical evidence is required to substantiate this claim.

Version 1:

Reviewer comments:

Reviewer #1

(Remarks to the Author)

The authors addressed my concerns properly, I am satisfied with the revisions.

Reviewer #2

(Remarks to the Author)

The authors have made significant revisions to address my previous comments, and these revisions have improved the manuscript considerably. However, I still find several statements difficult to follow, and there remain some issues of conceptual clarity, and unsupported claims that need to be addressed before the manuscript can be considered fully ready for publication.

1. Subtraction approach. The introduction describes creativity, semantic memory, and mental imagery as distinct but interconnected constructs, referencing the dual-coding theory, which posits distinct verbal and imagery-based representational systems. It is therefore unclear why semantic processing is treated as a confounding variable in the investigation of the neural basis of creativity–imagery interaction, rather than another potential (distinct) route to creativity. This inconsistency should be addressed, the rationale for controlling for the brain activity underlying creativity with a semantic strategy should be explicitly stated, along with the implications this has for interpreting the findings.

2. It appears the authors suggest that mental imagery contributes to cognitive control, as implied in the following statements:

- “Mental imagery participates in semantic memory processing by heteromodal recruitment and flexibly selecting task-relevant features.”
- “Our hypothesis posits that mental imagery facilitates creative cognition through two cognitive mechanisms within semantic memory processing: (1) semantic integration, which enables the binding of multimodal features into coherent conceptual representations, and (2) semantic reorganization, which dynamically restructures conceptual relationships to adapt to task demands.”

Such claims do not align with current models of semantic cognition or with the authors’ own description of mental imagery as inputs to a heteromodal semantic hub. The semantic control system is functionally and topographically distinct from the semantic representation (“hub-and-spoke”) system. A more accurate framing would be that modality-specific mental imagery inputs are modulated by top-down control processes in a task-dependent manner.

3. Several statements remain vague or confusing and require further explanation:

3.1. “Mental imagery participates in semantic memory processing by heteromodal recruitment...” and “Our results further support the hypothesis that mental imagery contributes to creative cognition by facilitating semantic memory processing through heteromodal recruitment...” What exactly is meant by “heteromodal recruitment”? Do you mean recruitment of heteromodal regions? If so, this contradicts the idea that mental imagery provides modality-specific inputs.

3.2. “Multimodal imagery is thus recruited and integrated into an amodal hub that integrates concepts.” What is meant by “integrates concepts”?

3.3. “Mental imagery serves as dynamic modality-specific inputs... engaging in integration across modules and is controlled by task relevance.” The term “modules” needs to be clarified. Do the authors mean modality-specific regions?

3.4. “The global supramodal hypothesis... modality-specific representations are inherently supramodal...” This statement is contradictory. Modality-specificity and supramodality are mutually exclusive.

3.5. “Highly creative individuals possess a relatively flat associative hierarchy...” The term “associative hierarchy” is

undefined. For clarity, explain that semantic memory can be conceptualized as a network of nodes (concepts), where edge weights reflect semantic similarity/conceptual associations. This will also help the readers follow the rest of the paragraph.

3.6. “These findings imply that functional brain organizations may not only featuring in non-overlapping brain modules.” This sentence is unclear and the implication does not follow intuitively from the described findings.

3.7. “These results highlight the importance of edges in brain networks...” Edges have always been integral to brain network modeling. If the authors intend to highlight the usefulness of edge functional connectivity approaches, this should be explicitly stated.

4. The manuscript uses several related terms—supramodal, multimodal, heteromodal—but it is unclear whether the authors consider these to be synonymous or conceptually distinct. If they are meant to be interchangeable, one consistent term should be used throughout. If distinctions exist, these should be clearly defined early in the manuscript.

5. Several strong interpretations are made based on reverse inference, which remains a concern I raised in my prior review. In particular, statements linking network activity to attention or concept integration are not empirically supported by the current data, as those functions were not directly measured during the fMRI task. Please adopt more cautious, evidence-aligned language. For example:

- “Findings for DAN and SAL revealed complex attention modes necessarily for creative cognition.”
- “Collectively, the results of the LIMA suggest the region integrates nonverbal and verbal ideas to form coherent concepts during creative writing.”

6. Minor - The following repetition should be removed for conciseness: “Firstly, edge communities are the result of clustering based on the fluctuation patterns of edges, thus they do not construct how information flows on edges. Firstly, edge communities, constructed through clustering edge fluctuation patterns, reflect statistical covariation rather than directional information flow.”

Version 2:

Reviewer comments:

Reviewer #2

(Remarks to the Author)

I am satisfied with the authors' response to my previous comments, and I have just one final minor point for clarification. The use of the term “(non)-active” in the following statements is unclear. Since the analyses are based on neural activity as measured via BOLD signal, it is not evident what the authors mean by “active” versus “non-active” in this context. I suggest rephrasing these statements for clarity:

- “which explains the non-active activities of LIMB in the results.”
- “The active similar communication within the SMNB may reflect its specialized role in auditory input processing.”

April 14, 2025
Dr. Jasmine Pan
Associate Editor
Communications Biology

Re: COMMSBIO-24-6761

Dear Dr. Pan,

We sincerely thank you for the valuable and insightful comments on our manuscript, titled “Cognitive and neural mechanisms of mental imagery supporting creative cognition”, and the opportunity to revise and resubmit for further consideration to *Communications Biology*. Your suggestions have been extremely helpful in guiding the revisions and improving the overall quality of the manuscript. We have carefully addressed each comment and made the necessary revisions accordingly. Below, we provide point-by-point responses to the reviewers' comments. The corresponding manuscript sections follow each response, with revisions highlighted in red font and unchanged text in grey.

Thank you again for your time and consideration. We hope that our manuscript is now suitable for publication in *Communications Biology*.

Reviewer #1:

Are they novel and will they be of interest to others in the community and the wider field?

The part of this work that presents a more innovative approach is the fMRI study, in which edge-centric functional connectivity is deployed as a tool to understand neural correlates of mental imagery during creative writing tasks. The emphasis of the tactile domain is also interesting. Finally, employing semantic integration and percolation analyses of semantic memory networks is an elegant methodological approach that can be appreciated by the scholarly community (although used before in the field by Kenett – amply cited by the authors)

We thank the Reviewer for their assessment of the novelty of our ms. We now provide more details on the percolation analysis in the Introduction (p. 7):

“While semantic network robustness captures the flexible reorganization of concepts, directly measuring cognitive flexibility (Cosgrove et al., 2021; Kenett, Levy, et al., 2018; Rastelli et al., 2022).”

and Materials and Methods sections (p. 35):

“A set of thresholds was defined for percolation analysis of the semantic network. The threshold range was from 0 to 1 with a step size of 0.0125 (i.e., threshold resolution). Edges with weights less than the threshold were iteratively removed from the semantic network at thresholds with a step size of 0.0125, and the size of the giant component (the largest connected group of nodes) was calculated. If the size of the giant component was less than 3 at a certain threshold, the iteration stopped. Finally, the percolation integral was calculated for each participant.

$$PI = \int_{start_TH}^{end_TH} GC(x)dx = \sum_{TH-start_TH}^{end_TH} GC(TH) * TH_res$$

PI is the percolation integral, GC is the giant component, TH is the threshold value, start/end_TH are the initial and final threshold values, and TH_res is the threshold resolution.”

If the conclusions are not original, it would be helpful if you could provide relevant references. Is the work convincing, and if not, what further evidence would be required to strengthen the conclusions?

The authors declare, as a limitation of the findings: “Future research could benefit from investigating the role

of mental imagery in other forms of creative expression, such as visual arts or musical composition” (line 883). For this reason, I would like to suggest that the previous literature review includes some works that partially fill that gap, such as Wong SSH, Lim SWH (2017) Mental imagery boosts music compositional creativity. PLoS ONE 12(3): e0174009. <https://doi.org/10.1371/journal.pone.0174009>.

The inclusion of De Pisapia, N., Bacci, F., Parrott, D. et al. Brain networks for visual creativity: a functional connectivity study of planning a visual artwork. Sci Rep 6, 39185 (2016). <https://doi.org/10.1038/srep39185> could also provide further context for networks involved in creativity in visual arts, in the literature section in lines 127 and following (De Pisapia published several other contributions on this topic).

We sincerely thank the Reviewer for the insightful comments. We have expanded and added the relevant citations in the limitations (p. 27):

“Future research could benefit from investigating the role of mental imagery in other forms of creative expression, **such as visual arts or musical composition (Palmiero, Nori, et al., 2016; Wong & Lim, 2017).**”

and introduction sections (p. 9):

“Based on **creativity neuroscience** research, extensive neuroimaging studies have suggested that creative cognition involves the integration of multiple cognitive processes, expanding from unimodal to multimodal brain regions (Chen et al., 2019; Patil et al., 2021; Zhuang et al., 2023). **The interaction between the executive control network and the default mode network constitutes the fundamental architecture of creative cognition, elucidating how self-generated thought and cognitive control collaboratively drive its mechanisms (Beatty et al., 2014, 2016, 2019; Chen et al., 2025; De Pisapia et al., 2016; Kenett et al., 2024; Matheson et al., 2023).**”

On a more subjective note, do you feel that the paper will influence thinking in the field? Please feel free to raise any further questions and concerns about the paper.

The paper has the potential to influence further research. I don't have major concerns in that regard, but rather a suggestion so that the authors don't miss the opportunity for widening the audience. In the paper's bibliography there is evidence that the authors considered a wider interdisciplinary angle. For example, the inclusion of more considerations from the point of view of neurobiology could open up the discussion in fruitful directions. As an acknowledgment of Nature's commitment to welcome interdisciplinary voices, also briefly crediting phenomenology and previous philosophy as seminal in sensory studies would have been fruitful (I have appreciated the definition of touch through poetic means – lines 690-693).

To introduce a wider interdisciplinary angle, we have expanded our discussion of mental imagery by integrating both phenomenological perspectives and cognitive theory approaches. We introduced mental imagery phenomenologically in the opening of the Introduction. The Introduction also includes newly added theoretical frameworks and hypotheses about mental imagery's role in semantic memory processing. In the Discussion, we expanded our discussion from the perspective of cognitive theories, specifically addressing "the contributions of mental imagery to creative cognition." The corresponding contents appear in the following order:

(p. 3): “Mental imagery, a fundamental ability enabling us to plan, rehearse future events, re-examine the past, and even simulate unreal scenarios (Pearson & Kosslyn, 2013), has been central to discussions of mental function for thousands of years, from philosophers to psychologists and now neuroscientists. Mental imagery is an information processing process that occurs in working memory (Pearson et al., 2015). It involves retrieving sensory attributes stored in long-term memory and constructing neural representations that resemble the results of perceptual processes (Tian & Poeppel, 2012). Such images can reflect veridical memories (e.g., a loved one's face) or fantastical constructs (e.g., a green elephant) (Roth, 2007). Some of the best anecdotal accounts of imagery facilitating imaginative thought come from scientists. Albert Einstein described his thought process as the involvement of the voluntary reproduction and combination of clear images (Miller, 2012). This one example highlights the importance of mental imagery in fostering high-order cognition, such as creative cognition (Daniels-McGhee & Davis, 1994).”

(p. 5): “However, research also indicates that mental imagery underpins semantic memory (Binder & Desai, 2011; Hoenig et al., 2008; Miller, 2004). Mental imagery serves as dynamic modality-specific inputs to semantic memory processing, engaging in the integration across modules and is controlled by task relevance. The hub-and-spoke model posits that cross-modal interactions are mediated by a bilateral anterior temporal lobe (ATL) hub, which itself contains no semantic features but represents the semantic similarity among concepts (Lambon Ralph et al., 2017; Patterson et al., 2007; Xu et al., 2016). Multimodal imagery is thus recruited and integrated into an amodal hub that integrates concepts (Ekstrand et al., 2017). Similarly, the neuro-cognitive model of semantic memory proposed by Fernandino et al. (2016) supports the notion of hierarchical sensory-motor integration as an essential architectural feature of semantic memory. Additionally, empirical research underscores the utility of embodied strategies, such as action or body simulations, in verbal divergent thinking tasks (Gilhooly et al., 2007; Matheson & Kenett, 2021; Sargent et al., 2023). Neuroimaging evidence further revealed that imagery engagement in semantic cognition is contextually optimized. Studies found that when the linguistic context emphasizes action properties, greater activation is observed in brain regions relevant for coding action information (van Dam et al., 2012). Recent studies support the finding by revealing that visual feature matching tasks require a visually focused brain state, engaging control regions for goal maintenance and prioritization of relevant visual knowledge, which must be functionally distinct from heteromodal conceptual hubs yet tightly integrated with task-specific sensory areas (Wang, Krieger-Redwood, et al., 2024). Taken together, mental imagery participates in semantic memory processing by heteromodal recruitment and flexibly

selecting task relevant features.

Several additional semantic memory hypotheses also highlight the role of mental imagery, albeit with varying emphases. Binder et al. (2009; 2011) propose that semantic memory consists of both modality-specific and supramodal representations, with mental imagery playing a dynamic role in language comprehension. Their framework suggests that the involvement of mental imagery changes through a gradual abstraction process, where detailed simulations are needed for unfamiliar or infrequent concepts, while simulations become less detailed as familiarity and contextual support increase. The global supramodal hypothesis further posits that modality-specific representations are inherently supramodal, enabling semantic representations to be task-oriented and perception-independent (Calzavarini, 2021). In essence, mental imagery's role extends beyond episodic memory to critically support semantic memory processes.”

(p. 20): “The contributions of mental imagery to creative cognition can be further understood through its distinct cognitive facets. Grounded in the theoretical models proposed by Kosslyn and colleagues (Kosslyn, 1996; Kosslyn et al., 2006), visual mental imagery can be divided into two principal facets: Representation, involving the generation and maintenance of imagery, and Manipulation, involving the examination and alteration of imagery. Emerging evidence suggests these facets differentially scaffold creativity. The representation of imagery provides raw material for creative ideation. For example, Kaufman and Kaufman (2009) reported that blocked writers are more likely to report low levels of positive, constructive mental imagery, and a lower level of vividness in their current work-related mental imagery activity. Another study found that young children who engaged in visual-auditory mental imagery demonstrated greater musical compositional creativity than those in the control group (Wong & Lim, 2017). These findings indicate that mental imagery representation is related to creative performance. The manipulation of imagery enables restructuring of mental content. Pearson and Logie (2015) discovered that external manipulation of mental imagery facilitated reference frame selection and change during creative integration. Palmiero et al. (2015; 2016) found that visual creativity was positively correlated with image transformation imagery ability. Additionally, a study found that imagery manipulation training improved undergraduate dance students' domain-general flexible thinking performance (May et al., 2020). Therefore, mental imagery also supports creative cognition through its manipulation facet. However, the current study did not differentiate between imagery representation and manipulation, and future research should investigate their distinct roles in creative cognition mediated by semantic cognition.”

We would also be grateful if you could comment on the appropriateness and validity of any statistical analysis, as well the ability of a researcher to reproduce the work, given the level of detail provided.

The level of detail in describing the methodology utilized seems quite comprehensive (which would also help with the replication of the experiments) the following observations (which relate more to the methods than the statistical analysis) need addressing:

- I would like to begin by commending the author for gathering such an extensive number of participants. However, the pool of subjects is comprised of majority female students in their 20s. A more varied and balanced population could have yielded different results, probably also based on variations in expertise, degree of studies and age.

We acknowledge that our participant pool may limit the generalizability of our findings. To address this concern, we conducted supplementary analyses for Study 1 (Figure S1 and Table S1). Our primary analyses remained robust even after statistically controlling for age and gender. Notably, in mediation analyses' Model 3, while the indirect effect approached conventional significance levels ($p = .0514$), the 95% CI [0.0008,0.0051] excluded zero, suggesting a marginally significant mediation effect.

(Supplementary Information p. 3): “**Fig. S2. Spearman's correlation heatmaps. (A) Spearman's correlations among the mental imagery vividness, creative writing scores, and semantic features. (B) Partial correlations controlling for age and gender.** The colors represent the magnitude of the r values, with black solid outlines indicating results that are significant after FDR (False Discovery Rate) correction with $p < .05$, and numbers denote the r values...

Table S1. Mediation analyses.

Model	Mediator	Outcome	Indirect (ab)	95% CI	Direct (c')	Total (c)
1	SI	CW_subjective	0.1150*	[0.0263,0.2611]	0.2960*	0.4110**
2	SNR	CW_subjective	0.2343**	[0.0929,0.4184]	0.1767	0.4110**
3	SI	CW_objective	0.0025	[0.0008,0.0051]	0.0071*	0.095**
4	SNR	CW_objective	0.0034*	[0.0011,0.0067]	0.0062	0.095**

Note. All models control for age, gender. Bootstrap samples = 1000. * $p < .05$, ** $p < .01$, *** $p < .001$.”

The supplementary analyses confirm demographic variables (age/gender) do not account for our core findings. However, we have expanded the Limitations section to include sample homogeneity, recommending future studies adopt lifespan sampling and writing professors.

(p. 27): “Our sample primarily consisted of college participants. Although we implemented statistical controls for age and gender, future investigations should aim to replicate these results in more diverse populations with broader demographic characteristics and varying levels of writing expertise (Erhard et al., 2014; Lotze et al., 2014).”

- *We don't know enough about the expertise of the subjects. Some may be training to become writers, or some may be using creative writing in the context of another creative profession, such as performing arts, visual arts or architecture (imagining and writing about the specificities of a movie set design, that gets built from the description, as an example). Such individuals may be using mental manipulation of semantic descriptions or images frequently in their practice. The role of expertise in creative tasks has been established, since expertise changes brain structures and functionalities in the specific brain areas that are used in a specific task [for examples, see Pascual-Leone, A. The brain that plays music and is changed by it. *Annals of the New York Academy of Sciences* 930, 315–329 (2001); Maguire, E. A., Woollett, K. & Spiers, H. J. London taxi drivers and bus drivers: a structural MRI and neuropsychological analysis. *Hippocampus* 16, 1091–1101 (2006)].*

Within the scope of the information we collected, we conducted supplementary analyses. We acknowledge that our data collection did not include attention to the expertise of the subjects, and we mention this in the Limitations section, as mentioned in our previous response.

(p. 27): “Our sample primarily consisted of college participants. Although we implemented statistical controls for age and gender, future investigations should aim to replicate these results in more diverse populations with broader demographic characteristics and varying levels of writing expertise (Erhard et al., 2014; Lotze et al., 2014).”

- *Lines 276-286 Task instruction seem very complex, especially for undergraduate students, with a high chance of non-compliance of execution due to the complexity of task. I suggest providing more details regarding the training phase to clarify how ambiguity due to poor task comprehension was avoided.*

In our main analyses, we didn't use data from the training phase. However, we recorded participants' ratings of mental imagery (MI) vividness and semantic understanding (SU) clarity for lemon-cutting, as well as their comprehension of the MI and SU concepts. To demonstrate that participants had a clear understanding of the task requirements, we have added the relevant descriptions in both the Materials and Methods section and the Results section, as well as modified Fig.2A.

(p. 30): “After the introduction, they rated, on a scale from 1 to 11, the vividness of their mental imagery and the clarity of their semantic understanding regarding lemon-cutting, as well as their comprehension of the concepts of mental imagery and semantic understanding.”

(p. 14): “Self-ratings of participants in instructional phase and across ten trials were analyzed on an 1-11 scale. Self-rated scores during the instructional phase averaged 9.01 ± 1.43 for mental imagery vividness and 9.76 ± 1.24 for semantic understanding clarity in lemon-cutting, and 9.34 ± 1.51 and 9.38 ± 1.38 for comprehension of mental imagery and semantic understanding concepts, respectively. One-sample t-test analyses confirmed these scores were significantly above the midpoint of 6 (all p's < .001), indicating participants clearly understood the task requirements (Fig. 2A).”

(p. 16): “Fig. 2. Creative writing performance under two conditions. (A) Average self-rated scores of participants in instructional phase and across ten trials.”

- Line 598 please correct the grammar – verb missing?

We sincerely thank the Reviewer for raising this point. We have now revised the sentence accordingly to address this issue.

(p. 15): “Semantic network robustness with added noise was significantly higher in the mental imagery condition than in the semantic understanding condition.”

- *What is the role of attention during task execution? Have you recorded any fluctuations and moments of loss of focus? It seems that 20 repetition of such a demanding task may generate decrease in focus and “creative fatigue”. Did you notice a difference in the first half of the set of tasks and the second half, in terms of performance by the each individual subject?*

The resource-control model of sustained attention posits that prolonged time-on-task depletes executive resources within SAL, which are essential for suppressing DMN activity and maintaining task focus [1], [2]. This suggests that mental fatigue typically manifests as enhanced DMN activation and reduced SAL engagement due to resource exhaustion. However, our findings revealed that both DMN and SAL nodes exhibited the lowest edge community similarity. This observation may indicate that SAL continued to function effectively during our experiment. Notably, our task-fMRI paradigm (30-minute duration) might not have been sufficient to induce SAL resource depletion. A study on compensatory neural activity [3] demonstrated that in cognitive control tasks, compensatory mechanisms remain active for up to 100 minutes before reaching the threshold for resource depletion. Different tasks have diverse compensation time threshold, but given this evidence, our task likely operated within the compensation phase, where SAL resources remained above depletion thresholds.

In addition, we revised our discussion regarding DAN and SAL. In the revised version, we highlight that creative cognition engages multiple attention mechanisms operating in tandem. Central to this process is SAL and DAN, which exhibit distinct yet complementary functional patterns for creative cognition.

(p. 23-24): “Findings for DAN and SAL revealed complex attention modes necessarily for creative cognition. We found the highest community overlap within the DAN and SAL networks, along with the highest edge community similarity in DAN_A but the lowest in SAL_A. Functionally, DAN is known to maintain goal-directed attention, whereas SAL detects behaviorally salient events and reorients attention (Corbetta et al., 2008; Menon & Uddin, 2010; Uddin, 2015; Vossel et al., 2014). The high overlap in DAN and SAL likely reflects their capacity to integrate goal-relevant signals and salient stimuli, as demonstrated in visual search (Geng & Mangun, 2011) and divergent thinking tasks (Ovando-Tellez, Benedek, et al., 2022), a critical process for creative thinking that requires both sustained focus on internal ideas and flexible capture of task-related cues from environment (Sun et al., 2019).

The highest edge community similarity in DAN_A suggest that it engages in highly coordinated information processing during the Thinking phase. This aligns with prior findings that DAN reduces its global connectivity (degree centrality) during goal-directed tasks (Kobayashi et al., 2020), possibly reflecting a shift from broad network integration to focal within-network synchronization. In our study, DAN_A's high edge community similarity may reflect it is potentially supporting stable attentional engagement. We found SAL_A showed the lowest edge community similarity. SAL is known to mediate switches between DMN and FPCN, suppressing task-irrelevant self-generated thoughts to guide task-relevant behavior (Denkova et al., 2019). SAL_A's broad

community distribution may reflect cooperation between brain networks associated with spontaneous thought and cognitive control (Beaty et al., 2015).”

Reviewer #2:

Major Comments

Introduction

1. The introduction includes key relevant content but is challenging to follow in places. To improve its structure and clarity, I recommend reorganizing the content to prioritize the hypotheses and their rationale before describing and motivating specific methods (e.g., eFC) or construct operationalizations (e.g., semantic integration vs reorganisation).

We sincerely thank the Reviewer for raising this important point. We have now adjusted the paragraph order of the introduction as follows: research objectives and concepts definitions - a review of studies on mental imagery and creativity - a review of studies on how mental imagery influences creativity through semantic memory and research hypotheses - to reveal the cognitive mechanisms, the research methods used in this study - a review of neuroscientific studies on mental imagery's involvement in creative cognition - a review of creative cognition studies at the brain network (node FC level) - a review of edge FC studies - summarizing research hypotheses and methods.

2. To improve the clarity of the theoretical grounding of the study. I recommend addressing the following:

• Define creative cognition in terms of its behavioral manifestations (e.g., painting, poetry, dance choreography) and its relevance to the hypotheses.

We now define creative cognition, based on the Reviewer's suggestion.

(p. 3-4): “This study investigates creative cognition via creative writing. The creative cognition approach views creativity as the generation of novel and appropriate products through the application of basic cognitive processes to existing knowledge structures (Ward, 2007). Ward (2001) established writing as a behavioral domain to study creative cognition, demonstrating how narrative construction inherently relies on cognitive processes like conceptual combination. Unlike traditional creative divergent or convergent thinking tasks,

creative writing entails a more intricate creative cognition process and reflects everyday creativity (creative hobbies, problem-solving in leisure or work activities) (D'Souza, 2021; Fürst & Grin, 2018; Hayes, 2000; Kaufman & Kaufman, 2009). Research has increasingly positioned creative writing as a behavioral manifestation of creative cognition. For instance, Taylor and Barbot (2024) demonstrated that the dual pathways of creativity (executive control and associative abilities) are directly applicable to creative writing productions, while studies on writing subprocesses reveal that optimal creativity emerges from dynamic patterns of idea generation and selection (Fürst et al., 2017). Notably, the sustained nature of creative writing could help to reveal the involvement of broader cognitive processes in creative cognition. Studies indicate that extended engagement with semantically complex and coherent narratives—as opposed to fragmented stimuli—recruits higher-order brain networks such as the default mode network (Lerner et al., 2011), suggesting that creative writing involves complex cognitive processes. Collectively, creative writing is a feasible way to reveal the involvement of broader cognitive processes in creative cognition.”

- *Clarify the distinction between modality-specific and supra-modal semantic representations and how they relate to creativity and mental imagery, as well as the brain networks referenced.*

We have now expanded our literature review on how mental imagery participates in semantic memory. Based on supramodal hypotheses and neuroimaging evidence, we propose that mental imagery serves as dynamic modality-specific inputs to semantic cognition, engaging in the integration across modules and is controlled by task relevance. The corresponding section is as follows. Notably, while we did not directly link modality-specific and supramodal semantic representations to creativity, our focus aligns with the broader research trends in the field of semantic creativity. Specifically, we emphasize the role of the structure of semantic memory and the dynamic search processes underlying creative cognition.

(p. 5-6): “However, research also indicates that mental imagery underpins semantic memory (Binder & Desai, 2011; Hoenig et al., 2008; Miller, 2004). Mental imagery serves as dynamic modality-specific inputs to semantic memory processing, engaging in the integration across modules and is controlled by task relevance. The hub-and-spoke model posits that cross-modal interactions are mediated by a bilateral anterior temporal lobe (ATL) hub, which itself contains no semantic features but represents the semantic similarity among concepts (Lambon Ralph et al., 2017; Patterson et al., 2007; Xu et al., 2016). Multimodal imagery is thus recruited and integrated into an amodal hub that integrates concepts (Ekstrand et al., 2017). Similarly, the neuro-cognitive model of semantic memory proposed by Fernandino et al. (2016) supports the notion of hierarchical sensory-motor integration as an essential architectural feature of semantic memory. Additionally, empirical research underscores the utility of embodied strategies, such as action or body simulations, in verbal divergent thinking tasks (Gilhooly et al., 2007; Matheson & Kenett, 2021; Sargent et al., 2023). Neuroimaging evidence further revealed that imagery engagement in semantic cognition is contextually optimized. Studies found that when the

linguistic context emphasizes action properties, greater activation is observed in brain regions relevant for coding action information (van Dam et al., 2012). Recent studies support the finding by revealing that visual feature matching tasks require a visually focused brain state, engaging control regions for goal maintenance and prioritization of relevant visual knowledge, which must be functionally distinct from heteromodal conceptual hubs yet tightly integrated with task-specific sensory areas (Wang, Krieger-Redwood, et al., 2024). Taken together, mental imagery participates in semantic memory processing by heteromodal recruitment and flexibly selecting task relevant features.

Several additional semantic memory hypotheses also highlight the role of mental imagery, albeit with varying emphases. Binder et al. (2009; 2011) propose that semantic memory consists of both modality-specific and supramodal representations, with mental imagery playing a dynamic role in language comprehension. Their framework suggests that the involvement of mental imagery changes through a gradual abstraction process, where detailed simulations are needed for unfamiliar or infrequent concepts, while simulations become less detailed as familiarity and contextual support increase. The global supramodal hypothesis further posits that modality-specific representations are inherently supramodal, enabling semantic representations to be task-oriented and perception-independent (Calzavarini, 2021). In essence, mental imagery's role extends beyond episodic memory to critically support semantic memory processes.”

- *Elaborate on what is meant by “creative cognition being embodied.” A deeper dive into the embodied cognition framework is needed to clarify the link between creativity, mental imagery, and semantic processing.*

We now thoroughly discuss “creative cognition being embodied” and the relationships between creativity, mental imagery, and semantic processing.

(p. 4-6): “Creative cognition is fundamentally embodied, with mental imagery serving as a critical cognitive process between sensorimotor experiences and creative outputs. This relationship manifests through two complementary research trajectories. First, embodied cognition studies highlight the scaffolding role of physicality in creative processes. Body movements and postures play a direct role in creative cognition (Frith et al., 2020; Hao et al., 2014; Sargent et al., 2023), potentially by enhancing the clarity of mental imagery – a mechanism proposed by Matheson and Kenett (2020) to explain how physical actions liberate creative thinking (see also Gilhooly et al., 2007). Neuroimaging evidence further corroborates this sensorimotor-creativity link: creative tasks consistently recruit functional coupling within sensorimotor brain regions (Kenett, Medaglia, et al., 2018; Matheson & Kenett, 2020; Ovando-Tellez, Kenett, et al., 2022; Sun et al., 2019), suggesting that mental imagery operates as an embodied cognitive strategy (Gilhooly et al., 2007; Matheson & Kenett, 2021; Sargent et al., 2023). Second, explicit investigations into mental imagery reveal its multidimensional

contributions to creativity. Researchers have examined the link between spatial visualization ability, self-reported visual imagery and creative performances (Palmiero et al., 2015; Palmiero, Nori, et al., 2016), respectively, and using imagery-based tasks to generate creative products (May et al., 2020; Wong & Lim, 2017). However, as Kozhevnikov et al. (2013) argued, such approaches often oversimplify both constructs, that creativity and mental imagery are not single homogeneous constructs but vary across domains. For instance, artistic versus scientific creativity differentially engage object versus spatial visualization abilities (Kozhevnikov et al., 2013; Palmiero et al., 2010). Thus, examining mental imagery in domain-specific creativity is essential.

The cognitive mechanisms of mental imagery supporting creative cognition likely operates through semantic memory, given its integral roles in prospection, mental imagery and divergent thinking (Abraham & Bubic, 2015). **Semantic memory stores facts, concepts, and general knowledge, which can be retrieved and combined in novel ways to foster creativity (Kumar, 2021).** Traditionally, mental imagery has been associated with episodic or visual memories (Farah, 1989; Hyman & Pentland, 1996; Keogh & Pearson, 2011; Kosslyn et al., 2001; Paivio, 1969). For instance, neuroimaging studies indicate that the brain regions involved in mental imagery overlap with those supporting episodic memory retrieval (Huijbers et al., 2011), which is logical since mental imagery often involves concrete, visualized memory aspects.

However, research also indicates that mental imagery underpins semantic memory (Binder & Desai, 2011; Hoenig et al., 2008; Miller, 2004). **Mental imagery serves as dynamic modality-specific inputs to semantic memory processing, engaging in the integration across modules and is controlled by task relevance.** The hub-and-spoke model posits that cross-modal interactions are mediated by a bilateral anterior temporal lobe (ATL) hub, which itself contains no semantic features but represents the semantic similarity among concepts (Lambon Ralph et al., 2017; Patterson et al., 2007; Xu et al., 2016). Multimodal imagery is thus recruited and integrated into an amodal hub that integrates concepts (Ekstrand et al., 2017). Similarly, the neuro-cognitive model of semantic memory proposed by Fernandino et al. (2016) supports the notion of hierarchical sensory-motor integration as an essential architectural feature of semantic memory. Additionally, empirical research underscores the utility of embodied strategies, such as action or body simulations, in verbal divergent thinking tasks (Gilhooly et al., 2007; Matheson & Kenett, 2021; Sargent et al., 2023). Neuroimaging evidence further revealed that imagery engagement in semantic cognition is contextually optimized. Studies found that when the linguistic context emphasizes action properties, greater activation is observed in brain regions relevant for coding action information (van Dam et al., 2012). Recent studies support the finding by revealing that visual feature matching tasks require a visually focused brain state, engaging control regions for goal maintenance and prioritization of relevant visual knowledge, which must be functionally distinct from heteromodal conceptual hubs yet tightly integrated with task-specific sensory areas (Wang, Krieger-Redwood, et al., 2024). Taken together, mental imagery participates in semantic memory processing by heteromodal recruitment and flexibly selecting task relevant features.

Several additional semantic memory hypotheses also highlight the role of mental imagery, albeit with varying

emphases. Binder et al. (2009; 2011) propose that semantic memory consists of both modality-specific and supramodal representations, with mental imagery playing a dynamic role in language comprehension. Their framework suggests that the involvement of mental imagery changes through a gradual abstraction process, where detailed simulations are needed for unfamiliar or infrequent concepts, while simulations become less detailed as familiarity and contextual support increase. The global supramodal hypothesis further posits that modality-specific representations are inherently supramodal, enabling semantic representations to be task-oriented and perception-independent (Calzavarini, 2021). In essence, mental imagery's role extends beyond episodic memory to critically support semantic memory processes.”

- *Provide definitions of the executive control network and the default mode network, including their subcomponents (e.g., FPCNa and FPCNb), and explicitly link them to the cognitive constructs of interest (creativity, mental imagery, and semantic memory).*

Regarding the definitions, we have elaborated on their primary roles in supporting creativity, including the specific contributions of the FPCNa subcomponent. These discussions are integrated into the relevant sections of the manuscript.

(p. 9): “The interaction between the executive control network and the default mode network constitutes the fundamental architecture of creative cognition, elucidating how self-generated thought and cognitive control collaboratively drive its mechanisms (Beatty et al., 2014, 2016, 2019; Chen et al., 2025; De Pisapia et al., 2016; Kenett et al., 2024; Matheson et al., 2023).”

(p. 9-10): “For example, Beatty et al. (2021) found distinct connectivity dynamics for subnetworks within the frontoparietal control network (FPCN) with the default mode network. Specifically, FPCNa (including rostral lateral prefrontal cortex, superior frontal gyrus, and anterior intraparietal sulcus) showed the greatest coassignment to the default network, reflecting dynamic reconfiguration during creative cognition.”

(p. 26): “Previous studies have shown that FCPN and DMN are central hubs in creative cognition by integrating complex information (Beatty et al., 2015; Chen et al., 2019; Zhuang et al., 2023). DMN enables complex forms of introspective thinking about stimuli not present in the environment, while FCPN helps individuals achieve goals by dynamically adjusting and switching attention and responses (Beatty et al., 2016; Konishi et al., 2015; Niendam et al., 2012). For instance, Kenett et al. (2020) examined the community structures of resting-state functional brain networks, and found that in higher-creative individuals, DMN regions have increased participation coefficients, suggesting a key role in coordinating diverse information across communities. **Finally,** recent studies highlight the role of dynamic and complex coupling between the DMN and FCPN that realize creative thinking (Chen et al., 2025; Kenett et al., 2024; Matheson et al., 2023), which align with current findings.”

3. *The rationale for the study's hypotheses and methodological choices would benefit from more clarity or expansion:*

- *The hypothesis that creative cognition supports “overlapping brain functional organization for information processing” lacks a clear rationale. Additionally, the importance of testing overlapping vs. non-overlapping functional architectures is not clear. Why is this question critical for the study, and for the field?*
- *The choice of edge-centric functional connectivity (eFC) as the primary analytical approach is unclear. Why is this method particularly suited to the study's research questions? Moreover, throughout the introduction, it was unclear to me whether the eFC would be derived from task-based fMRI or whether eFC metrics would be related to behavioral measures obtained outside the scanner.*

Since the first and second points address the rationale for using eFC and its relevance to the study, we combined our responses to them. Non-overlapping architectures indicate specialized node function for distinct cognitive demands, while overlapping architectures suggest complex neural activities within a single node. The edge-centric framework is particularly well-suited for studying creative cognition due to its ability to uncover overlapping functional organization. Creative cognition involves the dynamic integration of multiple cognitive processes, where a single network may participate in different types of information processing. However, the extent and mechanisms of this participation remain underexplored. We now fully discuss this in our revised ms.

(p. 10-11): **“Recently, an edge-centric framework was proposed to identify overlapping functional communities in brain networks, leveraging edge functional connectivity (eFC) to reveal previously obscured overlapping modular organization** (Betz et al., 2023). This recent method represents pairwise functional interactions between network edges (Faskowitz et al., 2021; Novelli & Razi, 2022). It involves computing the strength of functional connections between node pairs by unwrapping Pearson correlations over time, generating brain activity signals for each edge and observing fluctuations in their weights. Edge time series estimate edge-related structures, known as edge functional connectivity, which measures the similarity of time-varying co-fluctuations across the brain (Jo et al., 2021). High-amplitude eFC indicates strong similarity in communication patterns between edges, while low-amplitude eFC suggests independent fluctuations (Faskowitz et al., 2020). Clustering eFC classifies edges into non-overlapping edge communities. However, when these edge communities are mapped back to nodes, the partitions overlap, meaning a single node can be part of multiple edge communities (Ahn et al., 2010; Jo et al., 2021). **eFC extends and complements traditional node-centric brain network models, which measure inter-regional communication through temporal correlations. That is, strong functional connections are thought to reflect the time-averaged strength of communication between brain regions. eFC tracks how communication patterns evolve over time and ultimately assesses whether similar patterns are**

occurring in the brain simultaneously (Faskowitz et al., 2020). This approach provides a fine-grained perspective on brain network organization, capturing co-fluctuations in neural activity that may be critical for understanding complex cognitive processes.

Here, we utilized the edge-centric functional networks approach to analyze the synchrony of brain communication patterns, capturing dynamic brain organization during creative cognition with imagery encoding involvement. Recent studies found that edge-centric functional brain community distributions provide new insights into human cognition. A study found that edge communities vary across different brain system, such as heteromodal systems are more diverse than sensory systems and are more likely to form their own clusters (Jo et al., 2021). A recent study found brief periods of high-amplitude brain network connectivity in mice and humans, suggesting that these dynamic patterns are consistent across species (Ragone et al., 2024). Another study identified a general bridging factor in depression-anxiety comorbidity using edge-centric connectomes, revealing its robust generalizability (Chen et al., 2024). The edge-centric framework is particularly well-suited for studying creative cognition due to its ability to uncover overlapping functional organization. Creative cognition involves the dynamic integration of multiple cognitive processes supported by interactions between large-scale brain networks like the default mode network and executive control network (Beaty et al., 2019; Chen et al., 2025; Zhuang et al., 2021). eFC captures the moment-to-moment fluctuations in communication patterns, enabling the identification of overlapping functional communities that reflect the flexible reconfiguration of brain networks during creative tasks.”

- *The motivation for the operationalizations of “semantic integration” as semantic representation and “cognitive flexibility” as network robustness is not clear. Moreover, integration and representation are slightly different components of semantic processing and should not be equated.*

Thank you for pointing out the difference between semantic integration and representation, we have now changed representation into integration to avoid confusion.

(p. 7): “Our hypothesis posits that mental imagery facilitates creative cognition through two cognitive mechanisms within semantic memory processing: (1) *semantic integration*, which enables the binding of multimodal features into coherent conceptual representations, and (2) *semantic reorganization*, which dynamically restructures conceptual relationships to adapt to task demands. This framework is grounded in the roles of mental imagery in semantic processing, as well as the established role of semantic memory in creativity.

Creative writing offers a viable method for assessing semantic features. We propose quantifying the integration and reorganization of concepts within semantic memory using two semantic indicators derived from creative writing tasks. Semantic integration reflects the ability to connect distant associative concepts and weave rich ideas into cohesive narratives (Johnson et al., 2022), while semantic network robustness captures the flexible

reorganization of concepts, directly measuring cognitive flexibility (Cosgrove et al., 2021; Kenett, Levy, et al., 2018; Rastelli et al., 2022).”

4. *Some statements are vague, making it difficult to grasp the intended meaning:*

• *“Body movement and posture play a direct role in creative cognition.” Do you mean that many creative tasks involve bodily movement? It is unclear how whole-body movement relates to tasks like generating creative jokes. Providing specific examples could help clarify your argument.*

We have elaborated on the sentence in the revised manuscript by providing specific examples. The relevant explanation is now added to the ms as follows.

(p. 4): “Creative cognition is fundamentally embodied, with mental imagery serving as a critical cognitive process between sensorimotor experiences and creative outputs. This relationship manifests through two complementary research trajectories. First, embodied cognition studies highlight the scaffolding role of physicality in creative processes. Body movements and postures play a direct role in creative cognition (Frith et al., 2020; Hao et al., 2014; Sargent et al., 2023), potentially by enhancing the clarity of mental imagery – a mechanism proposed by Matheson and Kenett (2020) to explain how physical actions liberate creative thinking (see also Gilhooly et al., 2007). Neuroimaging evidence further corroborates this sensorimotor-creativity link: creative tasks consistently recruit functional coupling within sensorimotor brain regions (Kenett, Medaglia, et al., 2018; Matheson & Kenett, 2020; Ovando-Tellez, Kenett, et al., 2022; Sun et al., 2019), suggesting that mental imagery operates as an embodied cognitive strategy (Gilhooly et al., 2007; Matheson & Kenett, 2021; Sargent et al., 2023)…”

• *“We utilized the edge-centric functional networks approach to analyze blood oxygen level-dependent (BOLD) signal fluctuations of creative cognition with imagery encoding involvement to capture brain organization distribution patterns at a single-frame resolution.” What is meant by “BOLD of creative cognition”? Are you referring to functional connectivity during task-based fMRI? What is “imagery encoding involvement”? What do you mean by single frame resolution?*

We have now revised the sentence for clarity (please see below). The term “imagery encoding involvement” emphasizes the role of mental imagery in creative cognition, which is the experimental design in Study 3. By “single-frame resolution,” we refer to the edge-centric framework’s ability to reveal overlapping brain architectures based on edge co-fluctuations at each time point, in contrast to the node-centric framework, which relies on averaged fluctuations over time to assess regional similarities in brain activity.

(p. 11): “Here, we utilized the edge-centric functional networks approach to analyze the synchrony of brain communication patterns, capturing dynamic brain organization during creative cognition with imagery encoding involvement.”

Methods

5. The use of a subtraction approach in this study does not seem well-justified. The task conditions (semantic strategy and mental imagery strategy) appear to engage distinct cognitive processes rather than represent additive components of the same cognitive process. A more informative approach would have been to compare the characteristics of the brain networks estimated under each condition separately, highlighting the unique patterns associated with each strategy rather than relying on a subtraction-based comparison.

While we agree that the semantic understanding and mental imagery strategies may involve distinct cognitive processes, our choice of a subtraction approach was taken to isolate the neural characteristics that are relatively unique to the mental imagery strategy by controlling for potential confounding effects from the semantic strategy, such as language-related processing. The subtraction method allows us to focus on the overlapping features of brain communities of creative cognition especially with the involvement of mental imagery, which is the central research interest of the study. Since the investigation of the overlapping features of brain communities of creative cognition under semantic understanding conditions fall out of our research interest, we did not make corresponding adjustment. We have clarified this rationale in the revised ms.

(p. 37): “The time series of BOLD signals representing the creative writing process involving only imagery encoding was generated by concatenating the z-scored BOLD signals from 10 trials for 400 brain regions (Schaefer et al., 2018) under both conditions, then subtracting the semantic understanding condition from the mental imagery condition **to control for potential confounding effects from the semantic strategy (Fig. 5B).**”

6. The decision to set the number of communities a priori to 10 is not adequately justified. Why was this specific number chosen, and how does it align with the data or any theoretical considerations? To strengthen the analysis, the authors should provide evidence that $N=10$ is the optimal number of communities—such as through community quality metrics (e.g., modularity)—or demonstrate that their results are robust across a range of community numbers.

Thank you for raising this important point. We have now revised the selection of k in the main analysis. Following established methodological standards, we computed within-network entropy and similarity for each k from 2 to 20 and averaged the results to demonstrate the robust spatial distribution of edge communities.

Corresponding revisions were made in the Materials and Methods, Results, Discussion and Supplementary Information sections. In the Results, the previous description indicating that SMN_A showed the highest community overlap and $FCPN_A$ the lowest has been removed. We now add that TP exhibited the highest community similarity, while SMN_A and DMN_C showed the lowest. Relevant sections have also been updated accordingly.

(p. 39): “Following established methodological standards (Faskowitz et al., 2020; Jo et al., 2021), we applied k-means clustering with squared Euclidean distance (5000 iterations) to identify edge communities, varying the number of communities, k , from $k = 2$ to $k = 20$. Our main analyses report the average patterns across all k to demonstrate the robust spatial distribution of edge communities.”

(p. 22-23) “We found the lowest community similarity within SMN_A , and the highest community overlap and similarity within SMN_B . This suggests the SMN_A participated in variable communications patterns, whereas the SMN_B participated in much similar communication. The SMN_A consists of the dorsal and medial parts of anterior central gyrus (PreCG) and posterior central gyrus (ProCG). The SMN_B mainly includes ventral parts of PreCG and ProCG, terminating in the auditory cortex. According to the work of Kong et al. (2025), the SMN_A is associated with foot and hand somatomotor functions, whereas SMN_B is involved in auditory processing and face somatomotor activity. Numerous studies have shown the SMN's involvement in creative performances, reflecting imagery engagement (Bashwiner & Bacon, 2019; Bashwiner et al., 2016; Feng et al., 2019; Matheson & Kenett, 2020; Ovando-Tellez et al., 2023; Ovando-Tellez, Kenett, et al., 2022). The similar communication within the SMN_B may reflect its specialized role in auditory input processing or weak auditory imagery (Shergill et al., 2001). Our results indicate that nodes within SMN_A participated in various brain communication patterns, which reflect sensorimotor simulations in creative cognition. For a review, see Matheson and Kenett (2020). This supports the view that mental imagery operates as a dynamic information processing mechanism (rather than a static structural feature), driven by working memory's capacity to encode, maintain and manipulate sensory representations (Marvel et al., 2019).”

(p. 24-26) “Sequentially, the variability in community distribution among LIMA, LIMB, and TP may be related to their role of encoding associations between sensory stimuli and integrating them into ongoing cognitive operations in other parts of the brain during creative writing.

Firstly, LIM_A exhibited the highest edge community overlap and similarity. LIM_A encompasses the temporal pole and adjacent regions of the ventral anterior temporal lobe (vATL), which play a vital role in semantic cognition. Peelen and Caramazza (2012) proposed that conceptual object attributes exhibit more abstract along the posterior–anterior axis of the ATL, reaching peak representation in the temporal pole, further supported by Devereux et al. (2018). The vATL is thought to act as a semantic hub that combines verbal and nonverbal conceptual knowledge to form a semantic concept (Hoffman et al., 2014; Patterson et al., 2007; Ralph et al., 2017; Visser & Ralph, 2011; Xu et al., 2016). Studies indicate that ATL regions code semantic relationships between objects. An iEEG study observed that activity patterns of vATL were predicted by semantic similarity between objects (Chen et al., 2016). An fMRI study revealed that similar objects elicited similar patterns of

activation in the perirhinal cortex (the medial part of vATL) (Naspi et al., 2021). Similarly, the temporal pole is vital to semantic integration as proposed by a semantic hub model assuming that a semantic convergence zone in temporal pole for all kinds of concepts and semantics (Pulvermüller et al., 2010). The temporal pole is related to the acquisition of new conceptual knowledge (Hoffman et al., 2014), conceptual expansion (Abraham et al., 2012) and semantic categories processing (Pulvermüller et al., 2010). Studies found that the temporal pole contributes to creative thinking via semantic memory network (Yan et al., 2021). An fMRI study on creative writing observed a right lateralized activation network including the temporal pole (Shah et al., 2013). A study further found that experts in creative writing had reduced FC of the left caudate with left temporal pole, demonstrating that the disinhibition of the left temporal pole may contribute to excellent verbal creativity (Lotze et al., 2014). Collectively, the results of the LIM_A suggest the region integrates nonverbal and verbal ideas to form coherent concepts during creative writing.

LIM_B had the lowest community overlap and the highest community similarity, indicating that the activities in LIM_B during the task were less active and much singular. The orbitofrontal cortex (OFC, corresponding to LIM_B) receives highly processed sensory and emotion information and projects to the medial striatum, mediodorsal thalamus, and prefrontal regions (Rolls, 2004; Rudebeck & Rich, 2018), which enables the OFC to guide the efficient matching of behavior to environmental contingencies through flexible encoding (Rolls et al., 2005; Rudebeck & Rich, 2018). A clinical report had found the perfusion (hypoperfusion) in the OFC was significantly associated with lower creativity scores and the OFC may play a role in creativity via sensitivity to reinforcement and identification of emotions (De Souza et al., 2010). Structural neuroimaging studies suggest reduced OFC volume is associated with higher creative performance or divergent thinking abilities, indicating that thinner OFC is related to higher neural efficiency in supporting creative tasks (Jung et al., 2010; Zhang et al., 2021). Task-based fMRI studies on creative writing revealed that the brainstorming stage activated the bilateral OFC (BAs 10 and 11) (Shah et al., 2013), and the bilateral OFC (BA 10) was significantly activated when creative story generation was contrasted with uncreative story generation (Howard-Jones et al., 2005). The primate OFC was found to respond to novel but not familiar visual stimuli (Rolls et al., 2005). These findings indicate that the OFC may be associated with unique blot generation by directing to new stimuli for further processing (Gonen-Yaacovi et al., 2013). Time-window analyses further demonstrated that the OFC (BA 10) exhibited significant coupling with the DMN during later stages of alternate uses (vs. object characteristics), with no significant coupling in early stages (Beatty et al., 2015). Taken together, the OFC may contribute to creative cognition in a short temporal window by selectively responding to novel associations, which explains the non-active and singular activities of LIM_B in the results.

TP had the highest community similarity, which may indicate that it functions as a specialized site for multimodal convergence by receiving multisensory inputs (Shen et al., 2017). The superior temporal gyrus (STG) is heavily recruited during story/language comprehension (Turken & Dronkers, 2011; Virtue et al., 2006), novel associations (Shen et al., 2017) and original AUT evaluations (Maysseless et al., 2014). An fMRI study revealed that creative writing activated the posterior part of the left superior temporal region (Shah et al., 2013). TP is

located at the transition between the ventral and dorsal streams of visual processing while connects with the posterior DLPFC (Shen et al., 2017). The anatomical features of TP explain its functions of being the sites of semantic association and connected to the prefrontal cortex (Boccia et al., 2015).”

(Supplementary Information, p. 5-7):

“

Fig. S3. Violin plots depict the distribution of edge community overlap values across all nodes within the Yeo-Krienen 17-network parcellation, analyzed at multiple scales ($k = 2$ to 20).

Fig. S4. Violin plots depict the distribution of edge community similarity values across all nodes within the Yeo-Krienen 17-network parcellation, analyzed at multiple scales ($k = 2$ to 20).

Fig. S5. Sankey diagrams showing edge-to-node community relationships, with edge communities (left) linked to their nodes' Yeo-Krienen 17-network assignments (right) across multiple scales ($k = 2$ to 20).”

Results

7. The Results section includes unnecessary repetition of methodological details and would benefit from more concise descriptions.

We have now removed the methodological details from Study 2 and Study 3 in the Results section to improve clarity.

(p. 14): “Imagery strategy enhances creative writing performance

In Study 2, by comparing the creative writing performances under two encoding modes, i.e., non-verbal (imagery) and verbal coding, we examine whether mental imagery strategies could enhance creative writing performance...”

(p. 17): **“Brain edge community patterns under imagery encoding**

The analysis of edge community overlap during creative writing involving imagery encoding (mental imagery - semantic understanding condition) revealed that multiple edge communities were commonly present within a brain network, indicating parallel pathways for information communication within a brain network (Fig. 3A, 3B)...”

8. *The methods provided in the subsection titled “Imagery strategy enhances creative writing performance.” would be better suited in the Methods section.*

We have now removed the methodological details from this section to improve focus as stated in the previous answer.

9. *The description of self-reported measures is unclear. For example, the Likert scale extremes are not defined, making it difficult to interpret statements like: “One-sample t-test analysis revealed that mental imagery vividness and usage, as well as semantic understanding clarity and usage, were significantly above the average score of 6 (all p’s < .001).”*

We have now provided the scale information related to the instructional and formal writing phases in the Materials and Methods section (p. 30):

“After the introduction, they rated, on a scale from 1 to 11, the vividness of their mental imagery and the clarity of their semantic understanding regarding lemon-cutting, as well as their comprehension of the concepts of mental imagery and semantic understanding.”

(p. 30): “After writing, they rated the vividness of their mental imagery or the clarity of their semantic understanding for the cue words, as well as their application of the strategies in their writing, using a 1 to 11 scale.”

Discussion

10. *The results should be discussed with reference to the hypotheses stated in the introduction. As currently*

written, there is no reference to the hypotheses in the discussion.

To address this, we have added relevant references in the "The roles of mental imagery in creative cognition" Discussion section to strengthen the logical connection between our findings and the initial hypotheses. However, in the "Neural basis underlying creative cognition with imagery encoding" section, we did not add such references because the edge brain community analysis did not involve specific hypotheses.

(p. 19-20): “Our results further support the hypothesis that mental imagery contributes to creative cognition by facilitating the processing of semantic memory through heteromodal recruitment and the flexible selection of task-relevant features.”

(p. 21): “Our findings regarding the distinct patterns of semantic integration and semantic network robustness under the two conditions further indicate that these strategies differentially influence semantic memory processing.”

11. Several sentences in the discussion merely restate results without interpreting them and would benefit from conciseness. For example: “Furthermore, DANA showed both highest edge community overlap and similarity, indicating extensive similar information processing in DANA during creative writing.”

We have now revised the discussion about findings related to DAN_A. The highest edge community overlap in DAN_A indicates the capacity to integrate goal-relevant signals, while the highest edge community similarity in DAN_A indicates its stable attentional engagement.

(p. 23-24): “Findings for DAN and SAL revealed complex attention modes necessarily for creative cognition. We found the highest community overlap within the DAN and SAL networks, along with the highest edge community similarity in DANA but the lowest in SALA. Functionally, DAN is known to maintain goal-directed attention, whereas SAL detects behaviorally salient events and reorients attention (Corbetta et al., 2008; Menon & Uddin, 2010; Uddin, 2015; Vossel et al., 2014). The high overlap in DAN and SAL likely reflects their capacity to integrate goal-relevant signals and salient stimuli, as demonstrated in visual search (Geng & Mangun, 2011)…”

The highest edge community similarity in DAN_A suggest that it engages in highly coordinated information processing during the Thinking phase. This aligns with prior findings that DAN reduces its global connectivity (degree centrality) during goal-directed tasks (Kobayashi et al., 2020), possibly reflecting a shift from broad network integration to focal within-network synchronization. In our study, DAN_A's high edge community similarity may reflect it is potentially supporting stable attentional engagement.”

12. *Some interpretations are not fully supported by the data. For instance:*

- *“We further found the vividness of touch imagery influenced creative writing performance through two semantic features, i.e., semantic integration and semantic network robustness, which could be interpreted as vivid touch imagery promoting creative writing performance by representing novel concepts and flexibly manipulating these concepts.” No measure of haptic features of semantic representations or the novelty of concepts was explicitly assessed.*

We realized that the observed effects are related to semantic memory rather than the integration and reorganization of haptic features. Therefore, we have revised the corresponding discussion accordingly. We have clarified that the role of vivid touch imagery is to facilitate semantic memory processing, and our findings do not allow us to determine the extent to which the integration and reorganization of haptic features contribute to semantic memory processing.

(p. 19-20): *“Our results further support the hypothesis that mental imagery contributes to creative cognition by facilitating the processing of semantic memory through heteromodal recruitment and the flexible selection of task-relevant features. Specifically, vivid touch imagery enhances creative writing performance by promoting the integration of novel concepts and enabling their flexible manipulation within semantic memory. These findings align with previous hypotheses that semantic memory integrates sensory input into higher cognitive processes. For instance, embodied semantics theories propose that conceptual knowledge is grounded in sensory-perceptual experiences (Davis & Yee, 2021; Pulvermüller, 2013). In the information processing model of imagination proposed by Abraham and Bubic (2015), semantic memory plays a foundational role by abstracting content from specific sensory, motor, and affective experiences and engaging in broader semantic operations, such as creative thinking. Our results reveal that mental imagery contributes to creative cognition by facilitating semantic memory processing in two ways: by establishing connections to diverse semantic memory elements, and by flexibly manipulating connections between these elements.”*

- *“While both strategies lead to rich semantic representations, flexible manipulation of these concepts occurs only under the mental imagery condition” The authors did not provide direct evidence demonstrating that semantic reorganization is unique to the mental imagery condition, nor did they explicitly measure the richness of semantic representations.*

Upon reflection, we recognized a logical inconsistency in our explanation. Consequently, we have revised the sentences within the Discussion section and added in the Limitations section that semantic metrics serve as

an indirect measure of cognitive abilities, which does not equate to the comprehensive capacities of semantic integration and reorganization.

(p. 19): “While previous studies have not fully elucidated how touch affects cognition, it can be hypothesized that touch provides a more intimate and direct way of perceiving the world compared to other senses. Touch is central to many perceptual activities, forming a distinct yet overlapping touch memory that influences cognition and emotion (Gallace & Spence, 2020).”

(p. 27): “Secondly, semantic metrics provide a convenient, novel, objective, and ecologically valid way to reflect cognitive abilities, and previous studies have confirmed their capacity to reflect cognitive abilities (Gu et al., 2025; He et al., 2025; Johnson et al., 2022; Kenett, Levy, et al., 2018; Zhang et al., 2023). However, to verify the reliability of the results, future studies should incorporate cognitive tests to examine individuals' semantic integration and reorganization abilities.”

13. The interpretations that rely on speculative reasoning and reverse inference would benefit from the use of cautious, evidence-aligned language, such as “the results are consistent with the possibility that.” to more clearly differentiate between speculation and what the results demonstrate. For example, “The cooperation of both networks indicates that participants were able to flexibly switch their attention between internal and external resources to search for results, which support the dual-process model of creativity, positing an interaction (or coupling) between two latent cognitive modes (i.e., spontaneous/implicit thinking and deliberate/explicit thinking) during the creative cognition”. This interpretation is purely based on previous associations between the two networks and internal vs external attention. It is possible that those regions performed different functions during the tasks used in this study, and, therefore, the current results do not provide direct support for the dual process model of creativity.

We have now revised the discussion about findings related to DAN and SAL to avoid speculation.

(p. 23-24): “Findings for DAN and SAL revealed complex attention modes necessarily for creative cognition. We found the highest community overlap within the DAN and SAL networks, along with the highest edge community similarity in DAN_A but the lowest in SAL_A. Functionally, DAN is known to maintain goal-directed attention, whereas SAL detects behaviorally salient events and reorients attention (Corbetta et al., 2008; Menon & Uddin, 2010; Uddin, 2015; Vossel et al., 2014). The high overlap in DAN and SAL likely reflects their capacity to integrate goal-relevant signals and salient stimuli, as demonstrated in visual search (Geng & Mangun, 2011) and divergent thinking tasks (Ovando-Tellez, Benedek, et al., 2022), a critical process for creative thinking that requires both sustained focus on internal ideas and flexible capture of task-related cues from environment (Sun et al., 2019).

The highest edge community similarity in DAN_A suggest that it engages in highly coordinated information

processing during the Thinking phase. This aligns with prior findings that DAN reduces its global connectivity (degree centrality) during goal-directed tasks (Kobayashi et al., 2020), possibly reflecting a shift from broad network integration to focal within-network synchronization. In our study, DAN_A's high edge community similarity may reflect it is potentially supporting stable attentional engagement. We found SAL_A showed the lowest edge community similarity. SAL is known to mediate switches between DMN and FPCN, suppressing task-irrelevant self-generated thoughts to guide task-relevant behavior (Denkova et al., 2019). SAL_A's broad community distribution may reflect cooperation between brain networks associated with spontaneous thought and cognitive control (Beaty et al., 2015)."

14. *Some statements are vague, making interpretation difficult. For example:*

- *"DAN potentially executed creative writing tasks through internal attention processes involving substantial single information processing." What does "substantial single information processing" mean?*

We have revised the discussion about findings related to DAN (p. 23-24) as outlined above.

- *"Mental imagery engagement may function as an information processing mechanism, with working memory encoding sensory information to facilitate flexible information processing." This statement is too broad, as all cognitive functions could be said to involve "information processing mechanisms."*

We have revised the discussion about SMN to enhance clarity.

(p. 22-23) "We found the lowest community similarity within SMN_A, and the highest community overlap and similarity within SMN_B. This suggests the SMN_A participated in variable communications patterns, whereas the SMN_B participated in much similar communication. The SMN_A consists of the dorsal and medial parts of anterior central gyrus (PreCG) and posterior central gyrus (ProCG). The SMN_B mainly includes ventral parts of PreCG and ProCG, terminating in the auditory cortex. According to the work of Kong et al. (2025), the SMN_A is associated with foot and hand somatomotor functions, whereas SMN_B is involved in auditory processing and face somatomotor activity. Numerous studies have shown the SMN's involvement in creative performances, reflecting imagery engagement (Bashwiner & Bacon, 2019; Bashwiner et al., 2016; Feng et al., 2019; Matheson & Kenett, 2020; Ovando-Tellez et al., 2023; Ovando-Tellez, Kenett, et al., 2022). The similar communication within the SMN_B may reflect its specialized role in auditory input processing or weak auditory imagery (Shergill et al., 2001). Our results indicate that nodes within SMN_A participated in various brain communication patterns, which reflect sensorimotor simulations in creative cognition. For a review, see Matheson and Kenett (2020). This supports the view that mental imagery operates as a dynamic information processing mechanism (rather

than a static structural feature), driven by working memory's capacity to encode, maintain and manipulate sensory representations (Marvel et al., 2019).”

- “The highest edge community similarity indicates that the emotion processing within LIM_A was relatively singular.” What does “relatively singular” mean?

We sincerely thank the reviewer for the insightful comments. We have now revised the discussion about LIM_A. The new version explains the activity in LIM_A reflects its role of integrating nonverbal and verbal ideas to form coherent concepts during creative writing.

(p. 24-25): “Sequentially, the variability in community distribution among LIM_A, LIM_B, and TP may be related to their role of encoding associations between sensory stimuli and integrating them into ongoing cognitive operations in other parts of the brain during creative writing.

Firstly, LIM_A exhibited the highest edge community overlap and similarity. LIM_A encompasses the temporal pole and adjacent regions of the ventral anterior temporal lobe (vATL), which play a vital role in semantic cognition. Peelen and Caramazza (2012) proposed that conceptual object attributes exhibit more abstract along the posterior–anterior axis of the ATL, reaching peak representation in the temporal pole, further supported by Devereux et al. (2018). The vATL is thought to act as a semantic hub that combines verbal and nonverbal conceptual knowledge to form a semantic concept (Hoffman et al., 2014; Lambon Ralph et al., 2017; Patterson et al., 2007; Visser & Lambon Ralph, 2011; Xu et al., 2016). Studies indicate that ATL regions code semantic relationships between objects. An iEEG study observed that activity patterns of vATL were predicted by semantic similarity between objects (Chen et al., 2016). An fMRI study revealed that similar objects elicited similar patterns of activation in the perirhinal cortex (the medial part of vATL) (Naspi et al., 2021). Similarly, the temporal pole is vital to semantic integration as proposed by a semantic hub model assuming that a semantic convergence zone in temporal pole for all kinds of concepts and semantics (Pulvermüller et al., 2010). The temporal pole is related to the acquisition of new conceptual knowledge (Hoffman et al., 2014), conceptual expansion (Abraham et al., 2012) and semantic categories processing (Pulvermüller et al., 2010). Studies found that the temporal pole contributes to creative thinking via semantic memory network (Yan et al., 2021). An fMRI study on creative writing observed a right lateralized activation network including the temporal pole (Shah et al., 2013). A study further found that experts in creative writing had reduced FC of the left caudate with left temporal pole, demonstrating that the disinhibition of the left temporal pole may contribute to excellent verbal creativity (Lotze et al., 2014). Collectively, the results of the LIM_A suggest the region integrates nonverbal and verbal ideas to form coherent concepts during creative writing.”

Minor Comments

15. *“Which required participants to form sensory images based on given items and rate each image.” Specify that you mean mental images or images in the mind’s eye, rather than physical images, for precision.*

We have now made the corresponding modifications to address this issue.

(p. 29): “Mental imagery vividness was assessed using the Plymouth Sensory Imagery Questionnaire (Psi-Q) (Andrade et al., 2014), **which required participants to form mental imagery based on given items and rate each image** on a scale from 1 (no imagery at all) to 5 (imagery as vivid as reality).”

16. *Clarify that the participants completed the 20 creative writing tasks during fMRI scanning.*

We have now revised the description of the experimental design in Study 3 to improve clarity.

(p. 31-32): **“Participants completed 10 creative writing tasks for mental imagery and semantic understanding conditions, totaling 20 tasks (Fig. 5A). The order of the two conditions was randomized but fixed across participants. Event materials were divided into two sets: Set A consisted of the mental imagery version of 10 events and the semantic understanding version of another 10 events; Set B was the reverse.** The events mirrored those from Study 2, but text length was limited to 300 characters to minimize participant fatigue and distraction. Before fMRI scanning, the experimenter detailed the procedure, concept definitions, and requirements, highlighting the importance of focused attention and strategic writing over text summarization.”

17. *Expand on how the “200-dimensional word embeddings from Tencent AI Lab's Chinese term embedding corpus” were derived.*

We have now expanded the description of word embeddings.

(p. 33): **“Next, 200-dimensional word embeddings were obtained from the Tencent AI Lab Embedding Corpus for Chinese Words and Phrases (<https://ai.tencent.com/ailab/nlp/en/download.html>). This corpus includes 8.82 million words and phrases and was pre-trained using the Directional Skip-Gram (DSG) model (Song et al., 2018).”**

18. *“The cosine distance between each word in these two hidden layer representations was calculated.” Do you mean the cosine distance between each pair of words? Please clarify.*

We have now revised the sentence to accurately reflect this.

(p. 34): “The average cosine distance between each pair of words across the two hidden layer representations served as the semantic integration indicator for the entire story.”

19. Define the term “giant component” and explain what the “percolation integral (PI)” quantifies. Additionally, clarify how the existence or lack of a “dramatic change” in PI was determined or quantified.

We have now added the definition and formula of “giant component” and “percolation integral”, respectively.

(p. 35): “A set of thresholds was defined for percolation analysis of the semantic network. The threshold range was from 0 to 1 with a step size of 0.0125 (i.e., threshold resolution). Edges with weights less than the threshold were iteratively removed from the semantic network at thresholds with a step size of 0.0125, and the size of the giant component (the largest connected group of nodes) was calculated. If the size of the giant component was less than 3 at a certain threshold, the iteration stopped. Finally, the percolation integral was calculated for each participant.

$$PI = \int_{start_TH}^{end_TH} GC(x) dx = \sum_{TH=start_TH}^{end_TH} GC(TH) * TH_res$$

PI is the percolation integral, GC is the giant component, TH is the threshold value, start/end_TH are the initial and final threshold values, and TH_res is the threshold resolution.”

Regarding a “dramatic change” in the percolation integral (PI), we did not directly compare the PI values with and without added noise. Instead, we separately analyzed the PI of the semantic network under two conditions: (1) without added noise and (2) with added noise. In both conditions, the PI showed a significant correlation with creative writing ratings. This indicates that the relationship between semantic network robustness and creative writing ratings is stable. To avoid misunderstanding, we have modified the sentence.

(p. 35): “If the introduction of noise significantly alters the relationships between PI and other variables (e.g., creative writing ratings), it would indicate that the percolation process is not statistically robust.”

20. Avoid the use of bar graphs for presenting data, as they can obscure important details about distributions

(see Newman & Scholl, 2012).

In Study 2, we have now updated the results figure and adopted violin plots to more clearly visualize the comparison between the two conditions.

(p. 16): “Fig. 2. Creative writing performance under two conditions.”

21. “Community detection approaches applied to brain networks aim to find densely interconnected sets of nodes, leading to the notion that the brain is organized in a modular manner.” It is important to note that the ability to

apply these models to brain data does not necessarily imply that the brain functions modularly. Additional empirical evidence is required to substantiate this claim.

We have now modified the sentence as follows:

(p. 9): “Community detection approaches identify densely interconnected node modules, revealing how network organization supports cognitive functions.”

References:

- [1] P. Qi *et al.*, “Neural Mechanisms of Mental Fatigue Revisited: New Insights from the Brain Connectome,” *Engineering*, vol. 5, no. 2, pp. 276–286, Apr. 2019, doi: 10.1016/j.eng.2018.11.025.
- [2] D. Gui *et al.*, “Resting spontaneous activity in the default mode network predicts performance decline during prolonged attention workload,” *NeuroImage*, vol. 120, pp. 323–330, Oct. 2015, doi: 10.1016/j.neuroimage.2015.07.030.
- [3] C. Wang, A. Trongnetrpunya, I. B. H. Samuel, M. Ding, and B. M. Kluger, “Compensatory Neural Activity in Response to Cognitive Fatigue,” *J. Neurosci.*, vol. 36, no. 14, pp. 3919–3924, Apr. 2016, doi: 10.1523/JNEUROSCI.3652-15.2016.

June 16, 2025

Communications Biology

Re: COMMSBIO-24-6761

Dear Editor and Reviewers,

We sincerely thank you for the valuable and insightful comments on our manuscript, titled “Cognitive and neural mechanisms of mental imagery supporting creative cognition”, and the opportunity to revise and resubmit for further consideration to *Communications Biology*. Your suggestions have been extremely helpful in guiding the revisions and improving the overall quality of the manuscript. We have carefully addressed each comment and made the necessary revisions accordingly. Below, we provide point-by-point responses to the reviewers' comments. The corresponding manuscript sections follow each response, with revisions highlighted in red font and unchanged in grey font.

Thank you again for your time and consideration. We hope that our manuscript is now suitable for publication in *Communications Biology*.

Reviewers' comments:

Reviewer #1 (Remarks to the Author):

The authors addressed my concerns properly, I am satisfied with the revisions.

We thank the Reviewer for their praise of our revised ms.

Reviewer #2 (Remarks to the Author):

The authors have made significant revisions to address my previous comments, and these revisions have improved the manuscript considerably. However, I still find several statements difficult to follow, and there remain some issues of conceptual clarity, and unsupported claims that need to be addressed before the manuscript can be considered fully ready for publication.

We thank the Reviewer for acknowledging our efforts to revise the ms. We hope that continued efforts address all additional comments made below.

1. Subtraction approach. The introduction describes creativity, semantic memory, and mental imagery as distinct but interconnected constructs, referencing the dual-coding theory, which posits distinct verbal and imagery-based representational systems. It is therefore unclear why semantic processing is treated as a confounding variable in the investigation of the neural basis of creativity–imagery interaction, rather than another potential (distinct) route to creativity. This inconsistency should be addressed, the rationale for controlling for the brain activity underlying creativity with a semantic strategy should be explicitly stated, along with the implications this has for interpreting the findings.

We thank the Reviewer for pointing out the theoretical limitations of the analysis approach used in Study 3. In response, we have abandoned the subtraction approach and instead conducted separate analyses focusing on the characteristics of edge community distributions under the mental imagery and semantic understanding conditions. Accordingly, we have revised the Methods, Results, Discussion, and Supplementary Information sections to reflect these changes.

To sum up, findings of Study 3 demonstrate shared patterns of large-scale network engagement during creative writing under two conditions. Across both conditions, SMN_B plays a specialized role in auditory-related processing; DAN and SAL collaboratively support creative writing process by maintaining goal-directed attention and reorienting attention; LIM_A supports multimodal semantic processing; LIM_B may contribute to selectively respond to novel associations; FPCN and DMN support an interactive mechanism.

Meanwhile, two conditions demonstrate distinct patterns. Under the mental imagery condition, SMN_A facilitates sensorimotor simulations in creative cognition; DMN_B may play a special role in integrating semantic information related to objects and actions. In contrast, under the semantic understanding condition, LIM_A may reflect limited multimodal engagement or a tendency to process semantically similar information; TP participates in highly consistent semantic association processing; FPCN_C likely facilitates creative writing by updating working memory through integration of internal and external stimuli and prior knowledge to generate contextually coherent output.

2. *It appears the authors suggest that mental imagery contributes to cognitive control, as implied in the following statements:*

- *“Mental imagery participates in semantic memory processing by heteromodal recruitment and flexibly selecting task-relevant features.”*
- *“Our hypothesis posits that mental imagery facilitates creative cognition through two cognitive mechanisms within semantic memory processing: (1) semantic integration, which enables the binding of multimodal features into coherent conceptual representations, and (2) semantic reorganization, which dynamically restructures conceptual relationships to adapt to task demands.”*

Such claims do not align with current models of semantic cognition or with the authors’ own description of mental imagery as inputs to a heteromodal semantic hub. The semantic control system is functionally and topographically distinct from the semantic representation (“hub-and-spoke”) system. A more accurate framing would be that modality-specific mental imagery inputs are modulated by top-down control processes in a task-dependent manner.

We now highlight that mental imagery is influenced by top-down control processes. Specifically, we draw on the controlled semantic cognition framework proposed by Lambon-Ralph et al. (2017), which posits that the semantic representational system recruits multimodal inputs (both verbal and non-verbal) and is content-independent, while the semantic control system manipulates this representational system according to task demands. Based on this, we propose that mental imagery participates in semantic memory processing in a context-independent manner to support creative cognition. Mental imagery serves as modality-specific input that contributes to semantic integration which helps to form context-independently cohesive narratives. Meanwhile, mental imagery is involved in the flexible reorganization of semantic memory in a contextually optimized manner.

(p. 5-6) “However, research also indicates that mental imagery underpins semantic memory (Binder & Desai, 2011; Hoenig et al., 2008; Miller, 2004). **Mental imagery serves as modality-specific inputs to semantic representation and then integrates with semantic control processing (Lambon Ralph et al., 2017). Firstly, mental imagery contributes to semantic representation.** The hub-and-spoke model posits that cross-modal interactions are mediated by a bilateral anterior temporal lobe (ATL) hub, which itself contains no semantic features but represents the semantic similarity among concepts (Lambon Ralph et al., 2017; Patterson et al., 2007; Xu et al., 2016). Multimodal imagery, which refers to more than one sensory modality, is thus recruited and **interacts with a cross-modal hub to form generalizable concepts (Ekstrand et al., 2017; Lambon Ralph et al., 2017).** Similarly, the neuro-cognitive model of semantic memory proposed by Fernandino et al. (2016) supports the notion of hierarchical sensory-motor integration as an essential architectural feature of semantic memory. Additionally, empirical research underscores the utility of embodied strategies, such as action or body simulations, in verbal divergent thinking tasks (Gilhooly et al., 2007; Matheson & Kenett, 2021; Sargent et al., 2023). **Secondly,** neuroimaging evidence further revealed that imagery engagement in semantic cognition is contextually optimized. Studies found that when the linguistic context emphasizes action properties, greater activation is observed in brain regions relevant for coding action information (van Dam et al., 2012). Recent studies support the finding by revealing that visual feature matching tasks require a visually focused brain state, engaging control regions for goal maintenance and prioritization of relevant visual knowledge, which must be functionally distinct from heteromodal conceptual hubs yet tightly integrated with task-specific sensory areas (Wang, Krieger-Redwood, et al., 2024). **These findings align with the controlled semantic cognition framework, which proposes that top-down processes modulate activation within semantic representational systems to support goal-directed behavior (Lambon Ralph et al., 2017). Therefore, mental imagery participates in semantic memory processing and is manipulated by top-down control processes.”**

3. Several statements remain vague or confusing and require further explanation:

3.1. *“Mental imagery participates in semantic memory processing by heteromodal recruitment...” and “Our results further support the hypothesis that mental imagery contributes to creative cognition by facilitating semantic memory processing through heteromodal recruitment...” What exactly is meant by “heteromodal recruitment”? Do you mean recruitment of heteromodal regions? If so, this contradicts the idea that mental imagery provides modality-specific inputs.*

To maintain consistency with the above revisions, we have replaced the unclear expression (“heteromodal recruitment”) with the names of the two semantic indicators.

(p. 21) *“Our results further support the hypothesis that mental imagery contributes to creative cognition by facilitating the processing of semantic memory through **semantic integration and reorganization.**”*

3.2. *“Multimodal imagery is thus recruited and integrated into an amodal hub that integrates concepts.” What is meant by “integrates concepts”?*

To align with the theory (the hub-and-spoke framework), we have adopted terminology consistent with this framework. Regarding “integrates concepts”, it refers to the process by which modality-specific inputs are combined within the cross-modal hub to form context-independent concepts.

(p. 5) *“Multimodal imagery, which refers to more than one sensory modality, is thus recruited and **interacts with a cross-modal hub to form generalizable concepts (Ekstrand et al., 2017; Lambon Ralph et al., 2017).**”*

3.3. *“Mental imagery serves as dynamic modality-specific inputs... engaging in integration across modules and is controlled by task relevance.” The term “modules” needs to be clarified. Do the authors mean modality-specific regions?*

We have removed this expression and revised the wording to align with the terminology used in the controlled semantic cognition framework proposed by Lambon Ralph et al. (2017).

(p. 5) *“**Mental imagery serves as modality-specific inputs to semantic representation and then***

integrates with semantic control processing (Lambon Ralph et al., 2017).”

3.4. *“The global supramodal hypothesis... modality-specific representations are inherently supramodal...” This statement is contradictory. Modality-specificity and supramodality are mutually exclusive.*

We acknowledged that this statement is contradictory. Since the strength of the argument regarding mental imagery’s involvement in semantic memory processing in this paragraph is weaker than that of the preceding one, and its inclusion made the introduction overly lengthy, we removed this paragraph for the sake of conciseness.

3.5. *“Highly creative individuals possess a relatively flat associative hierarchy...” The term “associative hierarchy” is undefined. For clarity, explain that semantic memory can be conceptualized as a network of nodes (concepts), where edge weights reflect semantic similarity/conceptual associations. This will also help the readers follow the rest of the paragraph.*

We have added an explanation of the flat associative hierarchy and included a definition of the semantic network in the introduction.

(p. 6) *“According to the associative theory of creativity (Mednick, 1962), semantic memory structure reflects individual differences in creativity. Highly creative individuals possess a relatively flat associative hierarchy (numerous and weakly related associations to a given concept; e.g., Beaty et al., 2023; Hills & Kenett, 2025), allowing them to easily retrieve and combine remote associative elements. Modern computational network science tools have been applied to examine the structure and search process that operates over semantic memory related to creativity (Kenett, 2025; Kenett & Faust, 2019) by conceptualizing semantic memory structure as a network comprising nodes (semantic memory or concepts) and edges (semantic similarity)”*

3.6. *“These findings imply that functional brain organizations may not only featuring in non-overlapping brain modules.” This sentence is unclear and the implication does not follow intuitively from the described findings.*

We have refined the logical flow of this section by explaining how heteromodal networks support creativity through complex interactions, highlighting that a single network may be

involved in multiple types of information processing.

(p. 9-10) “There is evidence uncovering that brain organizations of creative cognition may be much more complex distributed. **Dynamic reconfiguration-based methods have revealed that heteromodal networks reconfigure their FCs and communities in various time windows during creative thinking (Beaty et al., 2015; Kenett et al., 2024; Lloyd-Cox et al., 2022; Patil et al., 2021; Sun et al., 2019), suggesting a single network can have varying network interactions during creative cognition. Further studies found that heteromodal networks collaborate but also demonstrate distinct FC patterns to support creative thinking and achievement (Sassenberg et al., 2025; Zhuang et al., 2023).** For example, Beaty et al. (2021) found distinct connectivity dynamics for subnetworks within the frontoparietal control network (FPCN) with the default mode network. Specifically, FPCNa (including rostrolateral prefrontal cortex, superior frontal gyrus, and anterior intraparietal sulcus) showed the greatest coassignment to the default network, reflecting dynamic reconfiguration during creative cognition. The complex roles of the FPCN in creative cognition are further revealed in a recent study (Wang, Chen, et al., 2024), which investigated the separate neural foundations of two creativity components, showing that the FPCN has less weight in both the novelty and appropriateness neural contributions. These findings imply that functional brain organizations **are highly complex, extending beyond a simple model of strictly segregated and non-overlapping brain modules** (Jo et al., 2021; Shankar et al., 2024), with a brain region playing multiple functional roles in creative cognition. However, the overlapping brain organization patterns of creative cognition, particularly during imagery encoding, have yet to be fully explored.”

3.7. *“These results highlight the importance of edges in brain networks...” Edges have always been integral to brain network modeling. If the authors intend to highlight the usefulness of edge functional connectivity approaches, this should be explicitly stated.*

We have removed this sentence to maintain conciseness and linguistic precision.

(p. 28-29) “In summary, findings of Study 3 demonstrate shared and distinct patterns of large-scale network engagement during creative writing under two conditions. Across both conditions, several networks exhibited consistent coherence: SMN_B plays a specialized role in auditory-related processing; DAN and SAL collaboratively support creative writing process by maintaining goal-directed attention and reorienting attention; LIM_A supports multimodal semantic processing; LIM_B may contribute to selectively respond to novel associations; FPCN and DMN support an interactive mechanism. Under mental imagery condition, SMN_A facilitates sensorimotor simulations in creative cognition; DMN_B may play a special role in integrating semantic information related to objects and actions. In contrast, under semantic understanding condition, LIM_A may reflect limited multimodal engagement or a tendency to process semantically similar information; TP participates in highly consistent semantic association processing; FPCN_C likely facilitates creative writing by updating working memory through integration of internal and external stimuli and prior knowledge to generate contextually coherent output.”

4. The manuscript uses several related terms—supramodal, multimodal, heteromodal—but it is unclear whether the authors consider these to be synonymous or conceptually distinct. If they are meant to be interchangeable, one consistent term should be used throughout. If distinctions exist, these should be clearly defined early in the manuscript.

Previously, we overlooked this issue. Since the paragraph containing “supramodal” has been removed, we have now added explanations of “multimodal” and “heteromodal” at the points where they first appear in the main text.

(p. 5) “Multimodal imagery, which refers to more than one sensory modality, is thus recruited and interacts with a cross-modal hub to form generalizable concepts (Ekstrand et al., 2017; Lambon Ralph et al., 2017).”

(p. 5) “Recent studies support the finding by revealing that visual feature matching tasks require a visually focused brain state, engaging control regions for goal maintenance and prioritization of relevant visual knowledge, which must be functionally distinct from heteromodal conceptual hubs (i.e., higher-order association cortices) yet tightly integrated with task-specific sensory areas.”

5. *Several strong interpretations are made based on reverse inference, which remains a concern I raised in my prior review. In particular, statements linking network activity to attention or concept integration are not empirically supported by the current data, as those functions were not directly measured during the fMRI task. Please adopt more cautious, evidence-aligned language. For example:*

- *“Findings for DAN and SAL revealed complex attention modes necessarily for creative cognition.”*
- *“Collectively, the results of the LIMA suggest the region integrates nonverbal and verbal ideas to form coherent concepts during creative writing.”*

Regarding DAN and SAL, we chose to adopt language that aligns more closely with previous research to describe their functions. Accordingly, we removed the original sentence from the discussion and replaced it with a more accurate phrasing.

(p. 29) *“DAN and SAL collaboratively support creative writing process by maintaining goal-directed attention and reorienting attention.”*

Regarding LIM_A, given that we no longer employed the subtraction approach, the corresponding discussion has been revised. We now report that LIM_A showed the highest edge community overlap under both conditions and exhibited the highest edge community similarity under semantic understanding condition. We interpret this as evidence that LIM_A may serve as a semantic representation hub. The differences in its edge community activity across the two conditions suggest that, although both conditions involved active communication patterns supporting multimodal semantic processing, the distinctive engagement of LIM_A under semantic understanding condition may reflect either reduced multimodal integration or a greater reliance on processing semantically similar information.

6. *Minor - The following repetition should be removed for conciseness: “Firstly, edge communities are the result of clustering based on the fluctuation patterns of edges, thus they do not construct how information flows on edges. Firstly, edge communities, constructed through clustering edge fluctuation patterns, reflect statistical covariation rather than directional information flow.”*

We have now removed the replicated sentence.

(p. 30) “Firstly, edge communities are the result of clustering based on the fluctuation patterns of edges, thus they do not construct how information flows on edges.”

June 23, 2025

Communications Biology

Re: COMMSBIO-24-6761B

Dear Editors and Reviewers,

We sincerely thank you for the valuable and insightful comments on our manuscript, titled “Cognitive and neural mechanisms of mental imagery supporting creative cognition”, and the opportunity to revise and resubmit for further consideration to *Communications Biology*. Your suggestions have been extremely helpful in guiding the revisions and improving the overall quality of the manuscript. We have carefully addressed each comment and made the necessary revisions accordingly. Below, we provide point-by-point responses to the reviewers' comments. The corresponding manuscript sections follow each response, with revisions highlighted in red font and unchanged in grey font.

Thank you again for your time and consideration. We hope that our manuscript is now suitable for publication in *Communications Biology*.

REVIEWERS' COMMENTS:

Reviewer #2 (Remarks to the Author):

I am satisfied with the authors' response to my previous comments, and I have just one final minor point for clarification.

The use of the term “(non)-active” in the following statements is unclear. Since the analyses are based on neural activity as measured via BOLD signal, it is not evident what the authors mean by “active” versus “non-active” in this context. I suggest rephrasing these statements for clarity:

- “which explains the non-active activities of LIMB in the results.”*
- “The active similar communication within the SMNB may reflect its specialized role in auditory input processing.”*

We thank the Reviewer for acknowledging our efforts to revise the ms. We hope that continued efforts address the final comment.

To maintain consistency with Jo et al. (2021), we described high within-network entropy as reflecting high uniformity in community assignments. We further defined high within-network similarity as a homogeneous community distribution to enhance conceptual clarity and avoid vagueness.

(p. 19) “This suggests the SMN_B participated in **uniform and homogeneous** communication under both conditions, whereas the SMN_A participated in variable communications patterns under mental imagery condition.”

(p. 20) “SAL_A's **heterogeneous** community distribution may reflect cooperation between brain networks associated with spontaneous thought and cognitive control¹¹⁰.”

(p. 21) “Taken together, although both conditions exhibited **high uniformity of community assignments** supporting multimodal semantic processing, the **homogeneous** activity of LIM_A under semantic understanding condition may reflect limited multimodal engagement or a tendency to process semantically similar information.”

(p. 21) “LIM_B had the lowest community overlap under both conditions, indicating that the

activities in LIM_B during the task were less **uniform**... Taken together, the OFC may contribute to creative cognition in a short temporal window by selectively responding to novel associations, which explains the **less uniformly distributed communities** of LIM_B in the results.”

References:

Jo, Y., Zamani Esfahlani, F., Faskowitz, J., Chumin, E. J., Sporns, O., & Betzel, R. F. (2021). The diversity and multiplexity of edge communities within and between brain systems. *Cell Reports*, 37(7), 110032. <https://doi.org/10.1016/j.celrep.2021.110032>